# Learning from Fine-Grained Visual Discrepancies: Mitigating Multimodal Hallucinations via In-Context Visual Contrastive Optimization

Haolin Deng [1 2]   Xin Zou [1 2]   Zhiwei Jin [3]   Chen Chen [3]   Haonan Lu [3]   Xuming Hu [1 2]

## Abstract

Multimodal hallucination remains a persistent challenge for Vision-Language Models (VLMs). Standard textual Direct Preference Optimization (DPO) often fails to mitigate it due to a lack of explicit visual supervision. While existing works introduce visual preference DPO by contrasting original images against negative ones, they suffer from a theoretically inconsistent objective caused by partition function mismatches and rely on coarse-grained negatives that could enable short-cut learning. In this work, we propose In-Context Visual Contrastive Optimization (IC-VCO). By placing contrastive images within a shared multi-image context, IC-VCO ensures a mathematically rigorous objective. We further introduce Visual Contrast Distillation (VCDist), an auxiliary reliability-gated regularizer that encourages consistency between multi-image contrastive training and single-image inference. Finally, we propose a contrastive sample editing strategy that generates hard negatives via precise semantic perturbations. Experiments on five benchmarks demonstrate IC-VCO's best overall performance and the effectiveness of our sample editing strategy. Code and data are available at https://github.com/OPPO-Mente-Lab/IC-VCO.

## 1. Introduction

Large Vision-Language Models (LVLMs) have demonstrated unprecedented capabilities in bridging visual perception and linguistic reasoning, revolutionizing tasks from visual question answering to embodied agency (Liu et al., 2023; Li et al., 2024; Bai et al., 2025b; Jin et al., 2025). To align these powerful models with human intent and mitigate toxic or untruthful generations, Direct Preference Optimization (DPO) (Rafailov et al., 2023) and reinforcement learning (Stiennon et al., 2020; Schulman et al., 2017; Guo et al., 2025) have emerged as two standard post-training paradigms. DPO, in particular, is favored for its stability and efficiency, as it does not require additional reward model training and policy roll-out during the training process.

Despite these advancements, LVLMs face a unique challenge: **Multimodal Hallucination** (Bai et al., 2024; Guan et al., 2024; Zou et al., 2025). Unlike textual hallucinations which often arise from factual (Min et al., 2023) or attribution errors (Deng et al., 2024), multimodal hallucinations frequently show a phenomenon called *visual neglect* (Wu et al., 2025b; Luo et al., 2025; Lu et al., 2022), where the model fails to ground its response in the provided visual input, relying instead on its language priors (Deletang et al., 2024). Standard DPO on textual preference treats the image merely as a static condition, so it often fails to effectively penalize the model for ignoring visual tokens. This limitation underscores the urgent need for optimization objectives that explicitly enforce visual grounding.

A growing line of multimodal preference optimization methods (Wang et al., 2024a; Xie et al., 2024; Yang et al., 2025; Wu et al., 2025b; Liu et al., 2025b) attempts to inject visual supervision into DPO-style training. One common approach is to construct visual preference pairs by fixing the textual response while contrasting a positive image against a negative one, and then optimize the standard DPO objective (Wang et al., 2024a; Yang et al., 2025; Wu et al., 2025b). The visual preference pairs can be constructed in a symmetrical way (Wu et al., 2025b). Specifically, given a standard image-response triplet $(m, x, y)$, where $m, x, y$ denote the input image, textual prompt and correct response respectively, a contrastive counterpart $(m', x, y')$ is introduced. $m'$ is curated to differ from $m$ in specific details. Consequently, the faithful response $y$ for $m$ contradicts the visual content of $m'$ (and vice versa for $y'$). This setup allows for the formulation of two symmetrical preference relations: $r(m, x, y) \succ r(m', x, y)$ and $r(m', x, y') \succ r(m, x, y')$, where $r$ denotes the implicit DPO reward. While effec-

[1]AI Thrust, the Hong Kong University of Science and Technology (Guangzhou) [2]The Hong Kong University of Science and Technology [3]OPPO AI Center. Correspondence to: Haolin Deng <hldeng028@gmail.com>, Xuming Hu <xuminghu@hkust-gz.edu.cn>.

*Proceedings of the 43^{rd} International Conference on Machine Learning*, Seoul, South Korea. PMLR 306, 2026. Copyright 2026 by the author(s).

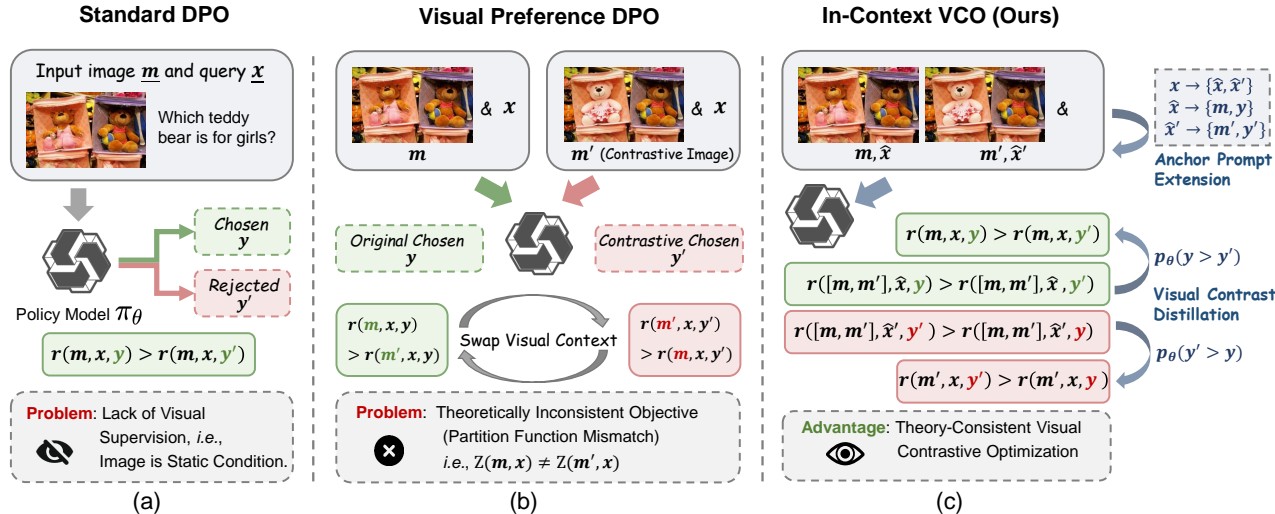

*Figure 1.* **Schematic comparison of preference optimization frameworks.** (a) **Standard DPO** optimizes textual preferences ($y$ *vs.* $y'$) while treating the image $m$ merely as a static condition, lacking explicit supervision for visual grounding. (b) **Visual Preference DPO** attempts to introduce visual rejected samples by changing visual context, *e.g.* swapping the input images ($m$ *vs.* $m'$). However, this approach suffers from a **theoretical inconsistency**: the partition functions $Z(m, x)$ and $Z(m', x)$ do not eliminate, leading to a non-rigorous optimization objective. (c) **In-Context VCO (Ours)** places both the original and contrastive images within a shared context $[m, m']$ and applies an anchor prompt extension step to specify the target image for preference labels. This design ensures a **theoretically rigorous** objective by sharing the partition function. A visual contrast distillation objective is introduced to calibrate the standard single-image DPO optimization with multi-image visual contrastive signals during simultaneous training.

tive for visual grounding, these approaches are limited by two theoretical and practical constraints:

❶ **Theoretically Inconsistent Objective.** While visual preference DPO methods effectively introduce contrastive visual signals to optimize implicit rewards, their objective functions rest on a loose theoretical approximation. By altering the visual contexts of preference pairs, the partition functions of the reference policy fail to cancel out. This results in a residual partition function ratio that persists as an intractable bias, rendering the optimization objective theoretically inconsistent with the original DPO formulation.

❷ **Coarse-Grained Negatives.** Recent works (Xie et al., 2024; Wu et al., 2025b; Liu et al., 2025b) typically construct contrastive images $m'$ via image retrieval or text-to-image synthesis. These images often exhibit distinct stylistic differences or noticeable semantic gaps compared to the original inputs. Such substantial deviations result in obvious and widespread inconsistencies between the image and contrastive response, making the rejected samples $(m, x, y')$ and $(m', x, y)$ *trivial negatives*: the model can minimize the DPO loss easily by exploiting the coarse-grained discrepancies, rather than learning the fine-grained visual facts.

In this work, we propose **In-Context Visual Contrastive**

**Optimization (IC-VCO)**, a framework which restructures visual alignment by placing contrastive images within a shared multi-image context. By instructing the model to distinguish and respond based on specific targeted images within this context, we ensure that the partition functions of preference pairs remain identical, strictly adhering to the theoretical consistency of DPO. Since IC-VCO constructs preference supervision in a multi-image context, while standard LVLM inference is performed with a single image, we further introduce **Visual Contrast Distillation (VCDist)** as an auxiliary consistency regularizer. VCDist uses the multi-image preference distribution as a soft reference to calibrate the single-image branch, encouraging the single-image policy to remain compatible with the contrastive training signal, thereby reducing the train-inference context gap.

To further address the challenge of trivial negatives prevalent in existing methods, we introduce a **Contrastive Sample Editing** strategy. Unlike previous approaches relying on retrieval or global synthesis which often introduce coarse stylistic discrepancies that facilitate shortcut learning, we employ a targeted editing pipeline. We perform precise, localized modifications on original images to generate high-quality hard negatives. These samples maintain strict stylistic consistency with the original visual distribution while embodying specific semantic contradictions, thereby compelling the model to develop fine-grained visual grounding capabilities rather than exploiting low-level short-

cuts. Comprehensive experiments on five diverse benchmarks demonstrate that IC-VCO provides a stronger multimodal preference optimization objective than existing baselines. The results also show that contrastive edited samples serve as broadly useful hard negatives, improving multiple preference-optimization methods beyond IC-VCO.

## 2. Preliminaries

### 2.1. Derivation of DPO Objective

DPO derives an analytical reward formulation from the generalized reinforcement learning objective:

$$r(x, y) = \beta \log \frac{\pi_\theta(y \mid x)}{\pi_{\text{ref}}(y \mid x)} + \beta \log Z(x), \quad (1)$$

$$Z(x) = \sum_y \pi_{\text{ref}}(y \mid x) \exp\left(\frac{1}{\beta} r(x, y)\right), \quad (2)$$

where $Z(x)$ is a partition function dependent on the input context and reference policy. The value of $Z(x)$ is intractable due to the implicit reward formulation. DPO assumes the reward model to be a Bradley-Terry model:

$$\begin{aligned} p(y_w \succ y_l) &= \frac{\exp\left(r(x, y_w)\right)}{\exp\left(r(x, y_w)\right) + \exp\left(r(x, y_l)\right)} \\ &= \sigma\left(r(x, y_w) - r(x, y_l)\right) \\ &= \sigma[\beta(\log \frac{\pi_\theta(y_w \mid x)}{\pi_{\text{ref}}(y_w \mid x)} + \log Z(x)) \\ &\quad - \beta(\log \frac{\pi_\theta(y_l \mid x)}{\pi_{\text{ref}}(y_l \mid x)} + \log Z(x))]. \end{aligned} \quad (3)$$

Eq. 3 shows that the chosen reward and rejected reward share the same value $Z(x)$, which can be eliminated. Thus, the DPO objective that tries to maximize the reward margin over preference pairs becomes:

$$\begin{aligned} \mathcal{L}_{\text{DPO}} = &-\mathbb{E}_{(x, y_w, y_l) \sim \mathcal{D}} \\ &\left[\log \sigma\left(\beta \log \frac{\pi_\theta(y_w \mid x)}{\pi_{\text{ref}}(y_w \mid x)} - \beta \log \frac{\pi_\theta(y_l \mid x)}{\pi_{\text{ref}}(y_l \mid x)}\right)\right]. \end{aligned} \quad (4)$$

### 2.2. Visual Preference DPO

Given an image $m$, textual prompt $x$, and a response pair $(y, y')$, where $r(m, x, y) \succ r(m, x, y')$, visual preference DPO (Wang et al., 2024a; Yang et al., 2025; Wu et al., 2025b) leverages a negative image $m'$ to optimize visual preference $r(m, x, y) \succ r(m', x, y)$ with DPO loss:

$$\begin{aligned} \mathcal{L}_{\text{VisDPO}} = &-\mathbb{E}_{(m, m', x, y) \sim \mathcal{D}} \\ &\left[\log \sigma\left(\beta \log \frac{\pi_\theta(y \mid m, x)}{\pi_{\text{ref}}(y \mid m, x)} - \beta \log \frac{\pi_\theta(y \mid m', x)}{\pi_{\text{ref}}(y \mid m', x)}\right)\right]. \end{aligned} \quad (5)$$

If $y'$ is the chosen response of $m'$, an extra visual preference pair $r(m', x, y') \succ r(m, x, y')$ can be leveraged for a

symmetrical loss (Wu et al., 2025b):

$$\begin{aligned} \mathcal{L}'_{\text{VisDPO}} = &-\mathbb{E}_{(m, m', x, y, y') \sim \mathcal{D}} \\ &\left[\log \sigma\left(\beta \log \frac{\pi_\theta(y' \mid m', x)}{\pi_{\text{ref}}(y' \mid m', x)} - \beta \log \frac{\pi_\theta(y' \mid m, x)}{\pi_{\text{ref}}(y' \mid m, x)}\right)\right]. \end{aligned} \quad (6)$$

The two losses can be jointly optimized.

**Issue: Theoretically Inconsistent Objective.** Given a visual preference pair $r(m, x, y) \succ r(m', x, y)$, following Eq. 3 and 4, we can derive the theoretical objective:

$$\begin{aligned} \mathcal{L}^*_{\text{VisDPO}} = &-\mathbb{E}_{(m, m', x, y, y') \sim \mathcal{D}}\left[\log \sigma\left(\beta \log \frac{\pi_\theta(y \mid m, x)}{\pi_{\text{ref}}(y \mid m, x)}\right.\right. \\ &\left.\left.- \beta \log \frac{\pi_\theta(y \mid m', x)}{\pi_{\text{ref}}(y \mid m', x)} + \beta \log \frac{Z(m, x)}{Z(m', x)}\right)\right]. \end{aligned} \quad (7)$$

By generalizing Eq. 2, we can get:

$$\log \frac{Z(m, x)}{Z(m', x)} = \log \frac{\sum_y \pi_{\text{ref}}(y \mid m, x) \exp\left(\frac{1}{\beta} r(m, x, y)\right)}{\sum_y \pi_{\text{ref}}(y \mid m', x) \exp\left(\frac{1}{\beta} r(m', x, y)\right)}. \quad (8)$$

Eq. 8 cannot be eliminated since $\pi_{\text{ref}}(\cdot \mid m, x)$ and $\pi_{\text{ref}}(\cdot \mid m', x)$ are different distributions. Eq. 5 and Eq. 6 essentially ignore this residual ratio, thereby optimizing a *biased proxy objective*. This residual term acts as a uncontrollable offset that shifts the implicit decision boundary arbitrarily for each training sample, ultimately restricting the optimization performance.

## 3. In-Context VCO

To address the theoretical inconsistency in Visual Preference DPO formulations, where distinct visual inputs lead to mismatched partition functions, we introduce a unified **In-Context Visual Contrastive Optimization (IC-VCO)** framework. Instead of processing images in isolation, we construct a shared *multi-image context* $M$ that encapsulates both the original image $m$ and the contrastive image $m'$. We define $M$ as a sequence of images $M = [m, m']$, containing both the original image $m$ and the contrastive negative $m'$, and feed them sequentially into the VLM. Given this multi-image context, the original textual prompt $x$ becomes ambiguous because the model cannot implicitly discern which image to ground its response on. To resolve this, we introduce an **anchor prompt extension** strategy to direct the model's attention to a specified target image within $M$.

Let $\hat{x}$ and $\hat{x}'$ denote the extended prompt targeting the original image $m$ and contrastive image $m'$ respectively. In practice, we explicitly append a positional anchor instruction to $x$ which is one of "*respond based on the first image*" or "*respond based on the second image*" depending on the

image order in $M$. To eliminate position bias in optimization, the image order is randomized for each sample. Based on this, we construct two symmetrical preference pairs: $r(M, \hat{x}, y) \succ r(M, \hat{x}, y')$ and $r(M, \hat{x}', y') \succ r(M, \hat{x}', y)$.

The multi-image objective for $r(M, \hat{x}, y) \succ r(M, \hat{x}, y')$ is:

$$p_{\text{multi}} = \sigma\left(\beta \log \frac{\pi_\theta(y \mid M, \hat{x})}{\pi_{\text{ref}}(y \mid M, \hat{x})} - \beta \log \frac{\pi_\theta(y' \mid M, \hat{x})}{\pi_{\text{ref}}(y' \mid M, \hat{x})}\right),$$
$$\mathcal{L}_{\text{Multi}} = -\mathbb{E}_{(M, \hat{x}, y, y')\sim\mathcal{D}}\left[\log p_{\text{multi}}\right]. \tag{9}$$

Here $p_{\text{multi}}$ is the chosen-response win probability under the shared context $(M, \hat{x})$. Since the chosen and rejected responses share the same condition, the partition function $Z(M, \hat{x})$ cancels out, yielding a theory-consistent objective.

Following prior multimodal preference optimization methods (Wang et al., 2024a; Yang et al., 2025; Wu et al., 2025b; Liu et al., 2025b), we also use single-image DPO:

$$p_{\text{single}} = \sigma\left(\beta \log \frac{\pi_\theta(y \mid m, x)}{\pi_{\text{ref}}(y \mid m, x)} - \beta \log \frac{\pi_\theta(y' \mid m, x)}{\pi_{\text{ref}}(y' \mid m, x)}\right),$$
$$\mathcal{L}_{\text{Single}} = -\mathbb{E}_{(m, x, y, y')\sim\mathcal{D}}\left[\log p_{\text{single}}\right]. \tag{10}$$

**Visual Contrast Distillation (VCDist).** IC-VCO constructs visual preference in a multi-image context $M$, whereas standard LVLM inference operates with a single image. To reduce the train-inference context gap, we introduce VCDist as an auxiliary consistency regularizer by using the multi-image preference distribution $p_{\text{multi}}$ as a soft reference to calibrate the single-image distribution $p_{\text{single}}$. This also encourages the single-image branch to absorb useful contrastive supervision from the multi-image context.

To ensure rigorous alignment, we introduce a *dual-gating mechanism* that filters the distillation signal based on correctness and relative confidence: a *correctness gate* filters out unreliable teacher signals (*i.e.*, $p_{\text{multi}} > 0.5$), while a *confidence gate* activates distillation only when the student's confidence falls below the teacher's (*i.e.*, $p_{\text{single}} < p_{\text{multi}}$) to prevent reverse penalty. Furthermore, to stabilize the optimization, we apply a *stop-gradient* operation to the teacher's logits. We formulate the VCDist objective as:

$$\mathcal{L}_{\text{VCDist}} = -\mathbb{E}_{(M, m, x, y, y')\sim\mathcal{D}}\Big[\mathbb{I}\big(p_{\text{multi}} > 0.5 \wedge p_{\text{single}} < \text{sg}(p_{\text{multi}})\big)$$
$$\cdot \Big(\text{sg}(p_{\text{multi}}) \log p_{\text{single}} + (1 - \text{sg}(p_{\text{multi}})) \log(1 - p_{\text{single}})\Big)\Big], \tag{11}$$

where $\mathbb{I}(\cdot)$ is the indicator function, and $\text{sg}(\cdot)$ denotes the stop-gradient operator.

Following previous works (Wang et al., 2024a; Yang et al., 2025; Liu et al., 2025b), we also leverage anchor losses to restrain the optimization from decreasing the chosen likelihoods compared to the reference policy. Since IC-VCO

contains both single-image and multi-image branches, we define the corresponding anchor terms separately:

$$\mathcal{L}_{\text{SingleAnc}} = -\mathbb{E}_{(m, x, y)\sim\mathcal{D}}\left[\log \sigma\left(\beta \log \frac{\pi_\theta(y \mid m, x)}{\pi_{\text{ref}}(y \mid m, x)}\right)\right], \tag{12}$$

$$\mathcal{L}_{\text{MultiAnc}} = -\mathbb{E}_{(M, \hat{x}, y)\sim\mathcal{D}}\left[\log \sigma\left(\beta \log \frac{\pi_\theta(y \mid M, \hat{x})}{\pi_{\text{ref}}(y \mid M, \hat{x})}\right)\right]. \tag{13}$$

The IC-VCO objective for the original image $m$ is defined by averaging the single-image and multi-image branches:

$$\mathcal{L}_{\text{IC-VCO}} = \frac{1}{2}\Bigg[\underbrace{\lambda_1(\mathcal{L}_{\text{Multi}} + \eta_1 \mathcal{L}_{\text{MultiAnc}})}_{\text{multi-image branch}}$$
$$+ \underbrace{\lambda_2(\mathcal{L}_{\text{Single}} + \eta_2 \mathcal{L}_{\text{SingleAnc}}) + \gamma \mathcal{L}_{\text{VCDist}}}_{\text{single-image branch}}\Bigg], \tag{14}$$

where $\lambda_1$, $\lambda_2$, $\eta_1$, $\eta_2$, and $\gamma$ are hyper-parameters. The anchor weights are coupled with corresponding preference branches, so that the $\mathcal{L}_{\text{SingleAnc}}$ is scaled together with $\mathcal{L}_{\text{Single}}$, while $\mathcal{L}_{\text{MultiAnc}}$ is scaled together with $\mathcal{L}_{\text{Multi}}$.

**Fine-grained Token-level Preference.** To further improve the single-image policy's sensitivity to fine-grained visual discrepancies, we introduce a token mask for the edited response pairs. Specifically, for response tokens that describe the edited visual evidence, we compute the single-image preference score only over the masked tokens. This token-level preference is applied to the single-image branch, while the multi-image branch still operates on the full response to preserve holistic visual reasoning.

**Symmetrical Preference Optimization.** Eq. 14 represents the objective when taking $m$ as target image, $y$ as chosen response, and $y'$ as rejected response. Since the contrastive sample $(m', x, y')$ is symmetrically constructed, following prior works (Wu et al., 2025b; Liu et al., 2025b), we can define a symmetrical objective $\mathcal{L}'_{\text{IC-VCO}}$ by taking $m'$ as the target image and $y'$ as the chosen response, similar to Eq. 6. The final objective can be expressed as:

$$\mathcal{L}_{\text{Total}} = \mathcal{L}_{\text{IC-VCO}} + \mathcal{L}'_{\text{IC-VCO}}. \tag{15}$$

## 4. Contrastive Sample Editing

To fully activate the theoretical potential of IC-VCO, we prioritize strict distributional alignment to prevent shortcut learning (Geirhos et al., 2020). From a fine-grained causal perspective (Pearl, 2009; Schölkopf et al., 2021), we decompose the image generation factors into three components: the *target semantic concept* $c_{tgt}$ (the focus of the preference pair), the *surrounding semantic context* $C_{ctx}$ (other visual

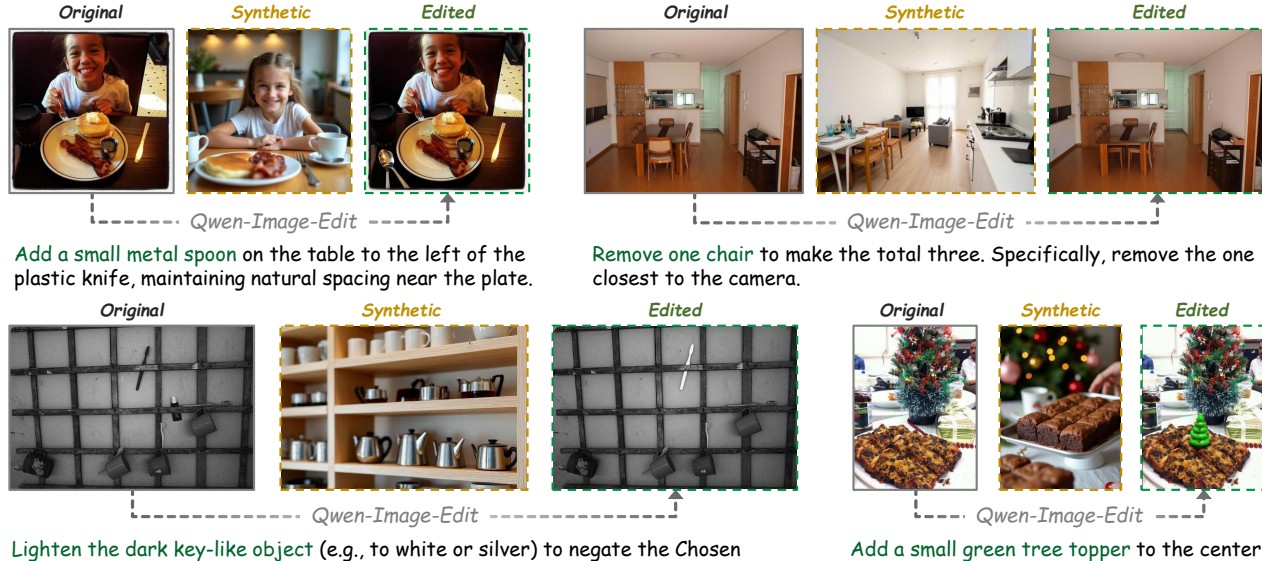

*Figure 2.* **Qualitative comparison of contrastive images.** Synthetic baselines (yellow) exhibit global stylistic shifts, acting as *coarse-grained negatives* prone to shortcut learning. In contrast, our Contrastive Editing (green) performs surgical, localized interventions while better preserving the visual context, yielding *fine-grained hard negatives* that compel rigorous visual discrimination.

entities), and *environmental factors U* (*e.g., style, lighting*). Thus, an image is modeled as $m = f(c_{tgt}, C_{ctx}, U)$.

Existing baselines typically construct negatives via *global resampling* processes that fail to preserve instance-level consistency. *Synthesis-based approaches* (Xie et al., 2024; Zhang et al., 2024; Wu et al., 2025b; Liu et al., 2025b), which utilize text-to-image diffusion models (Rombach et al., 2022), suffer from the *under-specification problem* (D'Amour et al., 2022): since textual descriptions cannot enumerate all background details, the model generates a new scene from the learned distribution, inadvertently altering the context. *Retrieval-based approaches* (Liu et al., 2025b) also introduce an independent realization of the scene by picking a distinct image $m'$ from a database. Both paradigms result in **coarse-grained negatives** exhibiting global semantic drift: $P(C_{ctx}, U|m) \neq P(C_{ctx}, U|m')$. Crucially, due to the *simplicity bias* of deep neural networks (Shah et al., 2020), these global distributional shifts provide a highly salient discriminative signal. Since distinguishing samples based on global style or background artifacts is easier than verifying fine-grained visual discrepancies, the model is more likely to collapse into a shortcut solution: rejecting $m'$ simply based on environmental inconsistencies rather than learning the target concept grounding.

In contrast, we define *hard negatives* as valid contrastive samples generated via surgical intervention: we aim to perform a precise operation $do(c_{tgt} \to c'_{tgt})$ while encouraging invariance on the semantic context and environmental factors: $\{C_{ctx}, U\}_m \approx \{C_{ctx}, U\}_{m'}$. This formulation ensures that the preference label primarily hinges on the

*Table 1.* Statistics of the contrastive sample editing dataset.

| Hallucination Type | Scenario A (Realization) | Scenario B (Injection) | Total |
|---|---|---|---|
| Attribute | 7,560 | 3,072 | 10,632 |
| Existence | 7,463 | 530 | 7,993 |
| Relation | 715 | 113 | 828 |
| **Total** | **15,738** | **3,715** | **19,453** |

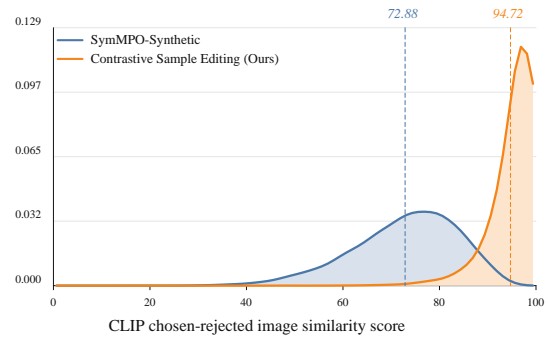

*Figure 3.* CLIP-based image pair similarity distribution.

fine-grained visual discrepancies of the target concept. To approximate this theoretical objective, we propose a **Contrastive Sample Editing Framework** utilizing a targeted edit pipeline to generate high-quality contrastive samples.

**Editing Pipeline.** Given a seed tuple $(m, x, y, y')$, we use QwenVL-Plus (Bai et al., 2025a) as an expert planner to generate an executable editing instruction $\mathcal{T}$. The planner handles two cases: *hallucination realization*, where an explicit hallucinated detail in $y'$ is made true in the edited

image $m'$, and *hallucination injection*, where a visual detail supporting $y$ is minimally contradicted when $y'$ does not provide a localized hallucination target. The target edit is categorized as *existence*, *attribute*, or *relation*. We then construct a contrastive response $y'_{\text{new}}$ by minimally rewriting the original chosen response $y$ according to $\mathcal{T}$, so that $(m', y'_{\text{new}})$ becomes the positive pair and $(m', y)$ becomes the negative pair while preserving the linguistic style of $y$. The token-level differences between $y$ and $y'_{\text{new}}$ are also recorded as fine-grained masks for token-level preference scoring. Next, Qwen-Image-Edit (Wu et al., 2025a) applies $\mathcal{T}$ to $m$ with a reversible padding procedure that preserves the original aspect ratio and maps the edited output back to the original resolution, keeping non-target regions aligned. Finally, QwenVL-Plus verifies whether $m'$ faithfully implements $\mathcal{T}$ and rectifies minor textual mismatches in $y'_{\text{new}}$; failed edits or samples with unintended structural changes are discarded. We provide full pipeline details in Appendix B.

**Data Source.** We use the **SymMPO dataset** (Liu et al., 2025b) as our seed corpus. This dataset comprises approximately 21.4k samples, originally aggregated from VQA v2 (Goyal et al., 2017), MSCOCO (Lin et al., 2014), and TextVQA (Singh et al., 2019) via the TPO dataset (He et al., 2024). Table 1 shows the core statistics of the contrastive sample editing dataset. We finally construct 19,453 contrastive samples, with an overall success rate of 91%.

**Quality Inspection.** Figure 2 compares synthetic and edited negatives against the original images. We also use CLIP (Radford et al., 2021) to calculate pair-wise image similarity scores. Figure 3 shows that edited images are more visually similar to the chosen images than synthetic images (mean CLIP image-similarity score: 94.72 vs 72.88). This further validates that our approach is effective at constructing hard negatives with fine-grained visual dependencies.

## 5. Experiment

### 5.1. Experimental Setup

**Training Data Statistics.** We use the SymMPO-synthetic dataset as our baseline training set. It contains 21.4k symmetrical image-text samples $(m, m', x, y, y')$ where the contrastive image $m'$ is generated via text-to-image synthesis. We also use our edited contrastive samples as a comparative training set with 19k symmetrical samples.

**Baselines & Architectures.** We implement our method on two representative open-source VLMs: LLaVA-NeXT-Interleave-Qwen-7B (Li et al., 2024) and LLaVA-OneVision-Qwen2-7B (Li et al., 2025a). We compare IC-VCO against the base models and leading multimodal preference alignment methods, including **mDPO** (Wang et al., 2024a), **V-DPO** (Xie et al., 2024), **S-VCO** (Wu et al.,

2025b), and **SymMPO** (Liu et al., 2025b)[1].

**Evaluation Benchmarks.** We employ five diverse benchmarks to comprehensively evaluate performance across different hallucination domains: 1) **HallusionBench** (Guan et al., 2024) assesses language hallucination and visual illusion capabilities. 2) **AMBER** (Wang et al., 2023) evaluates fine-grained hallucinations, specifically covering object existence, attributes, and spatial relations. 3) **CRPE** (Wang et al., 2024c) quantitatively tests object recognition and relation comprehension. 4) **R-Bench** (Li et al., 2025b) measures the model's robustness against various image corruptions and distortions. 5) **BLINK** (Fu et al., 2024) evaluates core visual perception abilities across 14 diverse computer vision tasks. To ensure the reproducibility of evaluation results, we use VLMEvalKit (Duan et al., 2024), an open-source evaluation toolkit to perform the standardized evaluations.

See Appendix C for implementation details.

### 5.2. Main Results

As presented in Tables 2 and 3, IC-VCO achieves the best overall score among all compared preference-optimization methods under both contrastive sample sources and both backbone models. On LLaVA-NeXT-Interleave, IC-VCO outperforms the strongest baseline by 1.19 points with synthetic samples and by 1.24 points with contrastive edited samples. On LLaVA-OneVision, IC-VCO surpasses the strongest baseline by 0.26 points and 0.48 points under the synthetic and edited settings, respectively.

The results also verify the general effectiveness of Contrastive Sample Editing. Replacing synthetic samples with edited samples shows improvements across baselines and base models. On LLaVA-NeXT-Interleave, the overall gains for the baselines range from 0.08 to 0.61 points, and IC-VCO improves from 62.83 to 63.35. On LLaVA-OneVision, the baseline gains range from 0.25 to 1.23 points, and IC-VCO improves from 66.26 to 66.82. These consistent improvements validate the advantage of fine-grained editing.

At the metric level, IC-VCO is particularly strong on attribute and existence-oriented grounding. It obtains the best AMBER-Attr score in all settings and also achieves strong AMBER-Exist and CRPE-Exist results. It also consistently improves BLINK, suggesting better general visual perception. However, relation-centric metrics are more mixed: IC-VCO does not always outperform all baselines on AMBER-Rel or CRPE-Rel. This may be partly due to the current edited dataset being skewed toward attribute and existence edits, with relation edits accounting for only 828 of 19,453 samples. We leave richer relation-oriented editing

---

[1]For fair comparison under the same annotation budget, we use a SymMPO variant without the extra $\mathcal{L}_{\text{DPO}_m}$ term that requires additional rejected-response annotations. See Appendix C.

*Table 2.* Experimental results on LLaVA-NeXT-Interleave-Qwen-7B. We compare the different methods using two distinct contrastive sample sources: Synthetic and Contrastive Sample Editing. The overall score denotes the macro-average across all benchmarks.

| Contrastive Sample Source | Approach | Overall | HallusionBench | | | AMBER | | | CRPE | | R-Bench | | BLINK |
| --- | --- | --- | --- | --- | --- | --- | --- | --- | --- | --- | --- | --- | --- |
| | | Score | aAcc | fAcc | qAcc | Attr | Exist | Rel | Exist | Rel | Dis | Ref | Score |
| | LLaVA-NeXT-Interleave-Qwen-7B | 59.14 | 55.59 | 25.76 | 25.90 | 79.97 | 89.03 | 74.51 | 92.01 | 60.20 | 55.96 | 59.11 | 45.13 |
| Synthetic (Liu et al., 2025b) | DPO(Rafailov et al., 2023) | 60.32 | 58.69 | 26.10 | 27.93 | 76.92 | 86.11 | 78.71 | 89.55 | 65.38 | 58.99 | 63.16 | 44.93 |
| | mDPO(Wang et al., 2024a) | 61.64 | 61.51 | 30.06 | 31.43 | 80.27 | 88.36 | 77.16 | 91.79 | 64.84 | 60.40 | 63.77 | 44.87 |
| | V-DPO(Xie et al., 2024) | 60.15 | 58.26 | 26.39 | 27.49 | 76.48 | 86.21 | 78.47 | 89.41 | 65.22 | 58.59 | 62.15 | 45.30 |
| | S-VCO(Wu et al., 2025b) | 60.81 | 58.68 | 27.17 | 28.79 | 77.88 | 86.17 | 79.21 | 89.92 | 66.00 | 59.80 | 63.36 | 45.19 |
| | SymMPO(Liu et al., 2025b) | 61.50 | 60.79 | 29.86 | 31.23 | 80.41 | 88.63 | 76.54 | 91.83 | 64.43 | 60.20 | 63.77 | 44.88 |
| | IC-VCO (Ours) | 62.83 | 61.94 | 30.82 | 31.55 | 81.81 | 90.48 | 75.56 | 93.16 | 65.63 | 59.70 | 63.87 | 48.93 |
| Contrastive Sample Editing (Ours) | DPO(Rafailov et al., 2023) | 60.40 | 60.46 | 27.17 | 30.11 | 78.16 | 92.47 | 76.20 | 89.35 | 63.31 | 57.58 | 61.34 | 44.66 |
| | mDPO(Wang et al., 2024a) | 62.02 | 60.25 | 29.48 | 30.99 | 80.31 | 92.55 | 74.64 | 92.27 | 65.36 | 60.00 | 65.79 | 45.66 |
| | V-DPO(Xie et al., 2024) | 60.38 | 59.94 | 26.88 | 30.99 | 77.77 | 91.98 | 76.50 | 89.20 | 62.88 | 57.98 | 61.34 | 44.87 |
| | S-VCO(Wu et al., 2025b) | 61.41 | 58.25 | 26.88 | 29.45 | 79.72 | 91.41 | 79.15 | 92.58 | 65.89 | 58.99 | 63.36 | 45.03 |
| | SymMPO(Liu et al., 2025b) | 62.11 | 60.57 | 29.77 | 31.65 | 80.39 | 92.89 | 74.52 | 92.47 | 65.58 | 61.01 | 65.18 | 45.19 |
| | IC-VCO (Ours) | 63.35 | 63.51 | 33.34 | 33.07 | 82.24 | 92.73 | 70.47 | 94.15 | 64.88 | 60.71 | 64.67 | 49.44 |

*Table 3.* Experimental results on LLaVA-OneVision-Qwen2-7B.

| Contrastive Sample Source | Approach | Overall | HallusionBench | | | AMBER | | | CRPE | | R-Bench | | BLINK |
| --- | --- | --- | --- | --- | --- | --- | --- | --- | --- | --- | --- | --- | --- |
| | | Score | aAcc | fAcc | qAcc | Attr | Exist | Rel | Exist | Rel | Dis | Ref | Score |
| | LLaVA-OneVision-Qwen2-7B | 62.46 | 53.84 | 25.04 | 24.95 | 84.05 | 91.67 | 75.98 | 94.52 | 65.26 | 66.16 | 71.96 | 44.82 |
| Synthetic (Liu et al., 2025b) | DPO(Rafailov et al., 2023) | 64.97 | 57.95 | 31.01 | 31.67 | 87.98 | 92.37 | 82.85 | 95.43 | 71.76 | 63.23 | 69.03 | 47.20 |
| | mDPO(Wang et al., 2024a) | 65.70 | 63.21 | 35.93 | 36.94 | 87.12 | 95.68 | 74.80 | 95.54 | 71.24 | 63.03 | 70.25 | 47.25 |
| | V-DPO(Xie et al., 2024) | 65.08 | 58.16 | 31.01 | 31.67 | 88.01 | 92.02 | 83.69 | 95.45 | 71.71 | 63.64 | 69.44 | 47.09 |
| | S-VCO(Wu et al., 2025b) | 66.00 | 62.26 | 36.51 | 36.94 | 88.04 | 92.94 | 81.71 | 95.30 | 71.18 | 63.44 | 69.64 | 47.41 |
| | SymMPO(Liu et al., 2025b) | 65.88 | 63.10 | 36.79 | 36.50 | 87.21 | 95.82 | 75.46 | 95.67 | 71.45 | 64.24 | 70.45 | 46.83 |
| | IC-VCO (Ours) | 66.26 | 62.15 | 35.55 | 36.26 | 88.06 | 95.98 | 73.20 | 95.94 | 71.75 | 65.25 | 71.05 | 48.92 |
| Contrastive Sample Editing (Ours) | DPO(Rafailov et al., 2023) | 66.08 | 63.34 | 37.06 | 37.88 | 86.01 | 94.24 | 80.99 | 94.94 | 71.53 | 64.05 | 68.43 | 47.75 |
| | mDPO(Wang et al., 2024a) | 66.24 | 63.90 | 36.36 | 37.45 | 86.11 | 96.28 | 76.39 | 95.61 | 71.93 | 64.50 | 70.10 | 47.99 |
| | V-DPO(Xie et al., 2024) | 66.31 | 63.90 | 37.80 | 38.55 | 86.42 | 94.65 | 81.56 | 95.46 | 72.19 | 64.70 | 68.88 | 46.63 |
| | S-VCO(Wu et al., 2025b) | 66.34 | 63.27 | 36.36 | 37.45 | 86.36 | 95.36 | 79.10 | 95.48 | 72.04 | 64.30 | 69.69 | 48.31 |
| | SymMPO(Liu et al., 2025b) | 66.13 | 64.00 | 36.07 | 37.67 | 86.06 | 96.24 | 76.03 | 95.48 | 71.83 | 63.89 | 70.30 | 47.89 |
| | IC-VCO (Ours) | 66.82 | 62.54 | 35.50 | 35.98 | 88.00 | 97.12 | 73.96 | 96.80 | 72.52 | 66.68 | 72.08 | 49.01 |

and supervision as future work.

### 5.3. Ablation Study

To validate the design of IC-VCO, we analyze component contributions in Table 4 and training diagnostics in Figure 4 on LLaVA-NeXT-Interleave-Qwen-7B. Compared to the full IC-VCO model on edited samples, removing the single-image branch decreases the overall score to 62.69, showing that single-image DPO is necessary for maintaining inference-time compatibility. Removing $\mathcal{L}_{\text{VCDist}}$ reduces the score to 63.04, indicating that calibrating the single-image branch with the multi-image preference signal is beneficial. Removing the token mask also hurts the overall score, which suggests that focusing the single-image preference signal on edited evidence tokens improves the effectiveness of fine-grained supervision. Removing the anchor loss $\mathcal{L}_{\text{Anc}}$ reduces overall score to 61.15, confirming that the anchor loss is important for stabilizing preference optimization by preventing the decline of chosen likelihoods.

**Multi-Image Branch and VCDist Signal.** Figure 4 (a) and (b) show that the multi-image branch consistently

achieves higher reward accuracy than the single-image branch. This validates the premise of VCDist: explicit visual comparison in the multi-image context provides a stronger preference signal. Figure 4 (c) further examines whether this stronger branch provides a reliable distillation target. The valid-teacher ratio measures the fraction of training samples for which the teacher passes the correctness gate. The valid KL measures the Bernoulli KL divergence between the teacher and student preference distributions on these valid samples. A stable valid-teacher ratio indicates that VCDist can continuously access a sufficient amount of teacher-aligned supervision, while the low valid KL suggests that the student remains close to the teacher distribution and the distillation target is well-conditioned rather than noisy or unstable. Together, these trends indicate that the VCDist signal is both active and reliable during training.

**Robustness of VCDist Design.** We further examine the stop-gradient and dual-gating mechanisms in Table 4. Removing stop-gradient slightly decreases the overall score from 63.35 to 63.12, while removing dual-gating slightly decreases it to 63.22. These mechanisms could be viewed as stabilization components rather than the sole source of per-

*Table 4.* Ablation study of IC-VCO. We analyze the impact of different components and the design choices within the VCDist.

| Method | Overall Score | HallusionBench | | | AMBER | | | CRPE | | R-Bench | | BLINK |
|---|---|---|---|---|---|---|---|---|---|---|---|---|
| | | aAcc | fAcc | qAcc | Attr | Exist | Rel | Exist | Rel | Dis | Ref | Score |
| IC-VCO | **63.35** | **63.51** | **33.34** | 33.07 | 82.24 | 92.73 | 70.47 | 94.15 | 64.88 | 60.71 | **64.67** | 49.44 |
| w/o single-image branch | 62.69 | 61.72 | 31.79 | 31.21 | 82.42 | 92.71 | 70.19 | **94.25** | 64.10 | 60.20 | 64.17 | 48.76 |
| w/o $\mathcal{L}_{\text{VCDist}}$ | 63.04 | 62.68 | 31.88 | 32.55 | 81.97 | 92.69 | 69.93 | 93.92 | 64.00 | 60.81 | 64.37 | 49.77 |
| w/o token mask | 63.10 | 61.72 | 31.02 | 31.53 | **82.43** | 92.32 | 72.58 | 94.00 | 64.36 | 60.30 | 64.27 | 50.18 |
| w/o $\mathcal{L}_{\text{Anc}}$ | 61.15 | 63.09 | 32.08 | **33.85** | 77.81 | 90.90 | **73.56** | 89.83 | 62.04 | 57.58 | 61.34 | 46.61 |
| *VCDist Ablation* | | | | | | | | | | | | |
| w/o stop-gradient | 63.12 | 62.67 | 33.05 | 32.19 | 82.30 | 92.99 | 70.29 | 93.97 | 64.41 | 59.90 | **64.67** | 49.65 |
| w/o dual-gating | 63.22 | 62.35 | 32.18 | 31.75 | 82.30 | **93.03** | 70.71 | 94.00 | **64.99** | 59.70 | **64.67** | **50.34** |

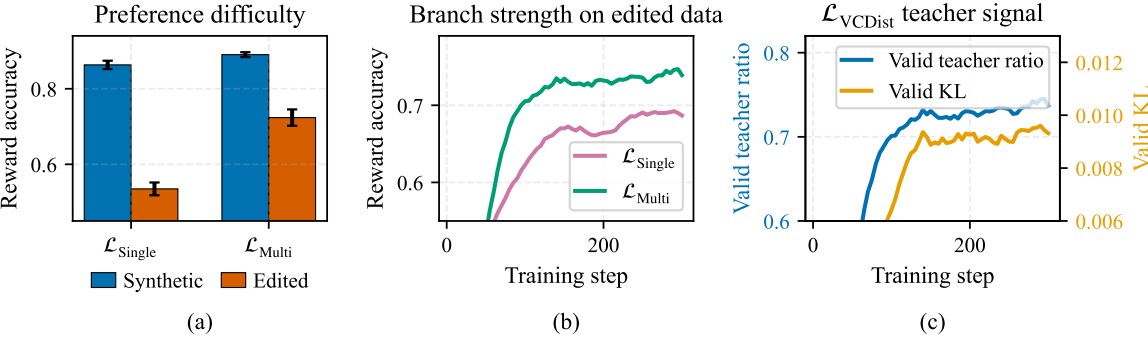

*Figure 4.* **Training diagnostics of IC-VCO on synthetic and edited preference data.** (a) Edited preferences are harder to optimize than synthetic preferences under the same response-level IC-VCO setup, yielding lower reward accuracy for both the single-image and multi-image branches. (b) On edited data, the multi-image branch consistently achieves higher reward accuracy than the single-image branch, indicating that multi-image comparison provides a stronger preference signal. (c) The VCDist objective maintains a stable valid-teacher ratio and low valid KL throughout training, showing that its teacher signal is active and well-conditioned.

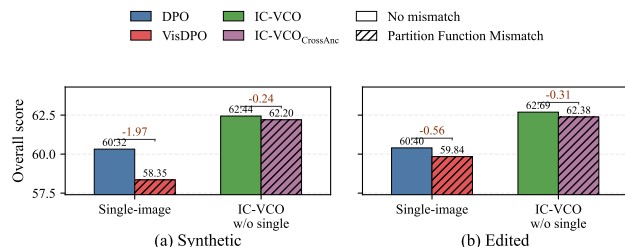

*Figure 5.* **Partition function bias analysis.** We remove the single-image branch of IC-VCO to form pure multi-image preference optimization. IC-VCO$_{\text{CrossAnc}}$ regroups the preference pairs by creating anchor prompt mismatch. The difference between DPO and VisDPO is shown in Eq. 4 and Eq. 5.

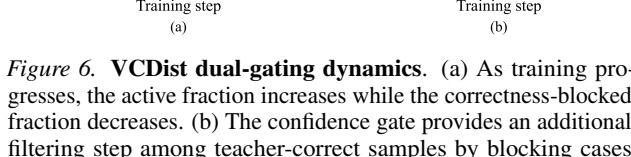

*Figure 6.* **VCDist dual-gating dynamics.** (a) As training progresses, the active fraction increases while the correctness-blocked fraction decreases. (b) The confidence gate provides an additional filtering step among teacher-correct samples by blocking cases where distillation is no longer needed.

formance gains: they make VCDist more conservative and robust without heavily changing the optimization objective.

### 5.4. More Analysis

**Sample Preference Difficulty.** Figure 4 (a) shows that the preference pairs with edited samples are harder to optimize than synthetic preferences, with lower reward accuracy for both the single-image and multi-image branches. This supports our claim that edited samples act as fine-grained hard negatives rather than trivial contrastive examples.

**Impact of Partition Function Mismatch.** Directly estimating $\log Z(m, x)/Z(m', x)$ is difficult in DPO because the reward is implicit and the partition term is policy-dependent over an open-ended response space. We therefore quantify its practical effect with a controlled ablation in Figure 5. In the single-image setting, the mismatch-based visual DPO variant underperforms the corresponding no-mismatch DPO baseline by 1.97 points on synthetic data and 0.56 points on edited data. In the multi-image setting, IC-VCO$_{\text{CrossAnc}}$ regroups preference pairs so that the same responses are compared across different anchor prompts, *i.e.*,

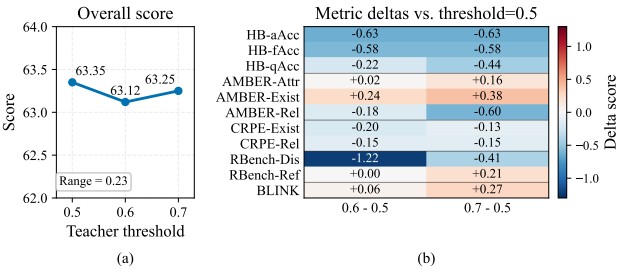

(a)          (b)

*Figure 7.* **Sensitivity analysis of the VCDist teacher threshold.** (a) Overall benchmark performance remains stable. (b) Per-metric differences show that the effect of threshold tuning does not lead to a consistent performance shift.

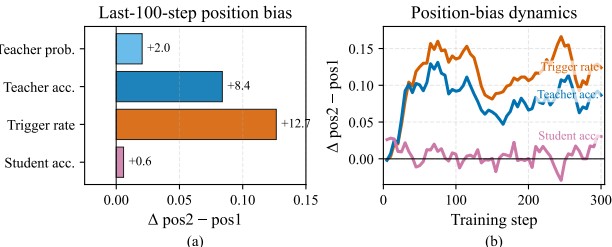

(a)          (b)

*Figure 8.* **Position-effect diagnostics for VCDist**. (a) The last-100-step mean deltas for the teacher probability, teacher accuracy, VCDist trigger rate, and student reward accuracy. (b) Representative deltas over training. The teacher-side statistics show a detectable positive position bias toward position 2, while the student single-image branch remains nearly position-symmetric.

$r(M, \hat{x}, y) > r(M, \hat{x}', y)$ and $r(M, \hat{x}', y') > r(M, \hat{x}, y')$, thereby reintroducing a context mismatch. This also hurts performance, but the degradation is smaller, possibly because the cross-anchor mismatch perturbs the conditioning much less than directly swapping images, and the multi-image visual contrast also provides stronger supervision, making the policy distribution more stable across anchors.

**VCDist Gate Dynamics.** Figure 6 analyzes how the VCDist gates behave during training. The dual-gating mechanism partitions training samples into active VCDist updates, confidence-blocked and correctness-blocked samples. The active fraction increases over training, while the correctness-blocked fraction decreases. This shows that the multi-image teacher becomes more reliable as training proceeds. The confidence gate is also empirically active: among teacher-correct samples, it filters 13.3% of cases on average in the last 100 steps. Overall, the dual-gating mechanism actively prevents undesired distillation while still allowing useful teacher signals to guide the single-image branch.

**Sensitivity of VCDist Threshold.** We test the sensitivity of the VCDist correctness threshold by changing it from 0.5 to 0.6 and 0.7. As shown in Figure 7, the overall score remains stable: the maximum change is only 0.23 points and per-metric differences do not show a consistent degradation pattern. This suggests that IC-VCO is not sensitive to the exact threshold choice within a reasonable range.

**Position Bias in Multi-Image Context.** Because IC-VCO uses positional anchor prompts, we further examine whether the multi-image teacher is affected by image order. We group training samples by whether the anchor-targeted image appears as the first or second image in the context. Figure 8 reports $\Delta = \text{Pos2} - \text{Pos1}$, where positive values indicate higher values when the anchor-targeted image is placed second. The teacher-side quantities show a detectable position-2 advantage: the last-100-step deltas are +2.0 points for teacher probability, +8.4 points for teacher accuracy, and +12.7 points for VCDist trigger rate. However,

the student reward accuracy changes by only +0.6 points, indicating that teacher-side asymmetry does not translate into a substantial single-image policy bias. Since the image order is randomized during training, the position effect is averaged across the dataset, while the dual-gating mechanism further restricts distillation to reliable teacher cases.

# 6. Related Work

To mitigate multimodal hallucinations, recent works (Wang et al., 2024a; Yang et al., 2025; Wu et al., 2025b) incorporate visual constraints into DPO by contrasting positive and negative image pairs. While pioneering, these approaches face two critical limitations. Theoretically, conditioning on distinct visual inputs prevents the cancellation of partition functions, violating the rigorous DPO derivation and introducing intractable bias. Practically, relying on retrieved or synthesized negatives often introduces coarse stylistic discrepancies (Liu et al., 2025b), creating *trivial negatives* that enable shortcut learning rather than enforcing fine-grained visual grounding. See Appendix A for detailed related works.

# 7. Conclusion

In this work, we present **In-Context Visual Contrastive Optimization (IC-VCO)**, a framework to address the theoretical inconsistencies in multimodal preference optimization for hallucination mitigation. By unifying contrastive images within a shared context, IC-VCO eliminates the intractable partition function bias, establishing a rigorous mathematical foundation for visual preference alignment. We also introduce **Visual Contrast Distillation (VCDist)**, a gated distillation regularizer to calibrate the single-image policy with the visual contrastive multi-image distribution. Furthermore, we propose a **Contrastive Sample Editing** pipeline which generates high-quality hard negatives to prevent shortcut learning on global distribution shift and enforce fine-grained visual grounding. Empirical results across diverse benchmarks validate the best overall performance of IC-VCO and the benefits from contrastive sample editing.

## Impact Statement

This work introduces a visual preference optimization framework for improving the reliability of Vision-Language Models. By encouraging models to distinguish fine-grained visual evidence through contrastive preference supervision, the method may help reduce visually inconsistent responses in applications such as visual assistants, educational tools, and content understanding systems.

However, our work should not be interpreted as a complete solution to multimodal hallucination or as a replacement for broader safety mechanisms. The method targets a specific post-training setting, namely DPO-style multimodal preference alignment, and is complementary to decoding-time, representation-editing, and human-feedback-based hallucination mitigation approaches. Models trained with our method may still produce incorrect or misleading outputs, especially in open-world or high-stakes scenarios such as medical diagnosis, legal evidence analysis, autonomous driving, or security surveillance, where additional validation and human oversight are necessary.

The proposed contrastive sample editing pipeline also relies on external expert VLMs and image editing models, which may introduce biases, verification errors, or editing artifacts. Moreover, edited counterfactual images could be misused if detached from their intended research context. We therefore recommend using the pipeline and samples only for controlled model training and evaluation. Finally, our approach introduces additional computational cost through multi-image training and model-assisted data construction. We release the materials to help reduce redundant generation costs for future research.

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

# A. Extended Related Work

## A.1. Multimodal Hallucination and Mitigation

Despite the remarkable capabilities of Large Vision-Language Models (LVLMs), they frequently exhibit multimodal hallucinations, where generated responses contradict visual content or fabricate non-existent objects (Bai et al., 2024; Guan et al., 2024; Luo et al., 2025). Recent analyses attribute this phenomenon to the model's over-reliance on language priors (Deletang et al., 2024; Luo et al., 2025) and insufficient visual grounding during the generation process (Lu et al., 2022; Xie et al., 2024; Wu et al., 2025b). Mitigation strategies can be broadly categorized into *training-free* and *training-based* approaches.

**Training-Free Approaches.** These methods intervene during inference or manipulate internal representations without updating model parameters. Decoding-based strategies, such as Visual Contrastive Decoding (VCD) (Leng et al., 2024), Instruction Contrastive Decoding (ICD) (Wang et al., 2024d), and VaLiD (Wang et al., 2024b), construct contrastive or modified decoding distributions to reduce hallucinations from different sources, including language priors and visual encoding distortions. Language-Contrastive Decoding (LCD) (Manevich & Tsarfaty, 2024) further refines this by directly contrasting VLM logits with those of a uni-modal LLM. MemVR (Zou et al., 2025) re-injects visual tokens before answer decoding to enhance visual grounding. Beyond decoding, representation editing methods (Liu et al., 2024; Jiang et al., 2025) manipulate activation patterns or suppress specific attention heads (Sarkar et al., 2025b; Ye et al., 2025) to enhance visual signal reliance. While effective, these methods often incur additional inference latency or require meticulous hyperparameter tuning for different architectures.

**Training-Based Approaches.** These methods align models via auxiliary visual supervision (Jiang et al., 2024; Sarkar et al., 2025a) or Reinforcement Learning (Stiennon et al., 2020; Yu et al., 2024a;b). Among them, Direct Preference Optimization (DPO) (Rafailov et al., 2023) based approaches have become a dominant paradigm due to its training stability and efficiency. Early multimodal adaptations, such as mDPO (Wang et al., 2024a) and V-DPO (Xie et al., 2024), extended textual DPO by optimizing preference between hallucinated and faithful captions while treating the image as a static condition. To explicitly enforce visual grounding, subsequent works like S-VCO (Wu et al., 2025b) and SymMPO (Liu et al., 2025b) introduced *visual contrastive optimization*, where the model learns to distinguish between matching and mismatched image-text pairs.

To explicitly enforce visual grounding, **mDPO** (Wang et al., 2024a) pioneers a visual preference optimization objective, constructing rejected samples by manipulating visual contexts to penalize image-text misalignment. To further mitigate the model's over-reliance on language priors, Xie et al. (2024) propose **V-DPO**, which incorporates a Vision-Guided Classifier-Free Guidance (CFG) framework also utilizing visual contrastive samples for optimization. Subsequently, Wu et al. (2025b) argue that utilizing preference pairs symmetrically—by assigning each response with a contradictory image—improves data efficiency. Their proposed **S-VCO** objective leverages contrastive image samples alongside text-only samples for visual preference modeling. Most recently, Liu et al. (2025b) identified that these prior visual DPO formulations are theoretically non-rigorous due to the partition function mismatch issue. As a theory-consistent alternative, they propose **SymMPO**, which restores DPO consistency by performing symmetric response-pair optimization under two separate single-image conditions $(m, x)$ and $(m', x)$, coupled by a preference-margin regularizer. In contrast, IC-VCO reformulates the task so that both contrastive images are placed in one shared context $M = [m, m']$ and the preference comparison is performed under the same full condition $(M, x)$. This allows IC-VCO to retain explicit in-context visual comparison while remaining theory-consistent.

Concurrent works have also explored fine-grained improvements to the DPO objective itself. For instance, TDPO (Zeng et al., 2024) proposes a token-level DPO formulation with forward KL divergence constraints; CHiP (Fu et al., 2025) designs a cross-model hierarchical DPO framework which combines textual preference at response-level, segment-level, and token-level. We note that these fine-grained preference optimizations are orthogonal to our research direction. In this work, we apply single-image token-level DPO using edited samples to all evaluated methods to align the granularity dimension. Furthermore, OPA-DPO (Yang et al., 2025) highlights the importance of on-policy data and aligns the preference data with the initial policy via finetuning. In our experiments, we apply this setting to all evaluated methods. MIA-DPO (Liu et al., 2025c) is similar to our IC-VCO as both works model multi-image DPO. The key difference lies in the supervision geometry. In MIA-DPO, the additional images are inserted as noisy distractors to induce multi-image hallucinations, the extra image mainly acts as a nuisance variable which is expected to be ignored by the model. In contrast, IC-VCO constructs semantically paired images which provide the supervision signal for explicit visual comparison while also supporting symmetrical optimization. MIA-DPO's approach could not support this, as the additional image is a distractor irrelevant to the question, which cannot ensure $r(M, \hat{x}, y') > r(M, \hat{x}, y)$. To further validate this, we sample 100 samples from the

MIA-DPO dataset[2] and use the expert VLM to evaluate the symmetry of preference pairs, and found only 21/100 pairs are symmetric. Additionally, MIA-DPO uses policy-coupled self-generated preference pairs, whereas IC-VCO operates on a fixed offline contrastive dataset shared across methods in our comparisons. IC-VCO also shows that jointly optimizing multi-image and single-image policies brings benefits, and proposes VCDist to bridge them.

### A.2. Visual Contrastive Data Construction

A pivotal challenge in multimodal preference optimization lies in constructing high-quality negative samples—specifically, contrastive images $m'$—that compel the model to attend to fine-grained visual details. Pioneering efforts (Wang et al., 2024a; Yang et al., 2025) typically derived $m'$ via heuristic augmentations, such as random cropping or noise injection. However, empirical studies (Wu et al., 2025b; Liu et al., 2025b) indicate that these corruption-based techniques yield suboptimal gains. Such drastic structural perturbations often destroy essential semantic information, rendering the negatives too easily distinguishable and prone to shortcut learning. Furthermore, these low-fidelity images prove ineffective under symmetric optimization frameworks, which require high-quality distributional matching.

To address this, subsequent approaches (Xie et al., 2024; Wu et al., 2025b; Liu et al., 2025a) have shifted towards leveraging text-to-image diffusion models (Rombach et al., 2022; Black Forest Labs et al., 2025) to synthesize distinct negatives conditioned on hallucinated captions, or utilizing image retrieval (Liu et al., 2025b) to source hard negatives. While domain-specific heuristics like horizontal flipping (Wu et al., 2025b) have also been explored for spatial relations, they lack generalizability. Crucially, as discussed in § 4, both synthesis-based and retrieval-based paradigms suffer from the *under-specification problem* (D'Amour et al., 2022), inevitably introducing global semantic drift. These deviations result in *coarse-grained negatives*, allowing models to bypass visual reasoning. In contrast, our Contrastive Sample Editing framework employs a localized sample editing strategy, implementing precise, surgical manipulations to generate *fine-grained negatives* that strictly preserve the surrounding context.

## B. Detailed Contrastive Sample Editing Pipeline

This section provides the detailed procedure used to construct the edited contrastive samples in Section 4. Given a seed preference tuple $(m, x, y, y')$, our goal is to produce an edited image $m'$ and a rewritten contrastive response $y'_{\text{new}}$ such that $(m, x, y)$ and $(m', x, y'_{\text{new}})$ form a symmetrical fine-grained contrastive pair, while $(m', x, y)$ becomes a hard negative whose error is localized to the edited visual evidence.

**Edit Strategy Formulation.** The first stage derives an executable image editing instruction $\mathcal{T}$ that introduces a precise semantic conflict between the edited image $m'$ and the original faithful response $y$. We use QwenVL-Plus (Bai et al., 2025a) as an expert VLM to analyze the seed tuple $(m, x, y, y')$ and formulate $\mathcal{T}$ under two scenarios.

- **Scenario A: Hallucination Realization.** If the rejected response $y'$ contains an explicit hallucinated detail that contradicts the original image $m$ and the faithful response $y$, the edit instruction $\mathcal{T}$ modifies $m$ to make one targeted hallucinated detail in $y'$ factually true in the edited image $m'$. After this intervention, the original chosen response $y$ naturally becomes a negative description for $m'$.

- **Scenario B: Hallucination Injection.** If $y'$ does not contain a localized hallucinated detail suitable for editing, we instead target the chosen response $y$. The edit instruction $\mathcal{T}$ minimally modifies $m$ to contradict a distinct visual detail described in $y$. In this case, the visual evidence supporting $y$ is surgically removed or altered, making $y$ a negative description for the edited image $m'$.

For both scenarios, the expert VLM classifies the target edit into one of three hallucination types: *existence*, *attribute*, and *relation*. We additionally enforce strict constraints on the editing instruction to avoid structural changes, global style shifts, or background modifications. This ensures that $m'$ remains visually congruent with $m$ except for the targeted semantic concept.

**Contrastive Response Rewriting.** To construct a valid symmetrical preference pair, we require a contrastive response $y'_{\text{new}}$ that faithfully describes the edited image $m'$. The original rejected response $y'$ is not directly used as the positive

---

*Table 5.* Computation overhead of visual editing: Qwen-Image-Edit-2511 with a fused Lightning LoRA in bf16, 8 inference steps per image, and the reversible padding/unpadding pipeline. Statistics are reported over 100 image-editing runs on a single NVIDIA H20 GPU.

| Inference steps per image | Measured images | Mean latency | Median latency | P90 latency | Peak GPU memory | Total time for 100 images |
|---|---|---|---|---|---|---|
| 8 | 100 | 14.02 s/image | 14.00 s/image | 14.08 s/image | 58.79 GiB | 1402.05 s (23.37 min) |

response for $m'$ because it may contain multiple hallucinations in Scenario A or may be unrelated to the selected edit target in Scenario B. Therefore, we adopt a minimal intervention rewriting strategy. Specifically, the expert VLM rewrites the original chosen response $y$ by changing only the keywords or short phrases necessary to align with the edit instruction $\mathcal{T}$, while preserving the sentence structure, reasoning pattern, and linguistic style of $y$.

This rewriting strategy has three benefits. First, it inherits the coherence and detailed reasoning of the original chosen response, avoiding the noise often present in rejected responses. Second, the resulting pair $(m', y'_{new})$ versus $(m', y)$ differs only in the atomic concept targeted by the edit, which prevents the model from exploiting textual style, length, or reasoning-format shortcuts. Third, because $y'_{new}$ is produced by minimally modifying $y$, we can directly compute the token-level differences between the two responses and extract fine-grained masks for token-level preference scoring.

**Fine-Grained Visual Editing.** We use Qwen-Image-Edit (Wu et al., 2025a) to apply the editing instruction $\mathcal{T}$ to the original image $m$. In our implementation, the editor is instantiated with the Qwen-Image-Edit-2511[3] base checkpoint and a fused Lightning LoRA checkpoint, Qwen-Image-Edit-2511-Lightning[4]. We fuse the LoRA with scale 1.0 in bf16, and use 8 denoising steps with true_cfg_scale=1. The editor input resolution is fixed to $1024 \times 1024$, the random seed is fixed to 42, and we use an empty negative prompt.

A practical challenge is that image editing models typically operate at a fixed square resolution. Naively resizing the original image to this resolution can distort the aspect ratio and break the spatial alignment between $m$ and $m'$. To avoid such artifacts, we adopt a reversible padding pipeline. We first resize $m$ to fit the $1024 \times 1024$ canvas while preserving its aspect ratio using Lanczos resampling, then pad the remaining margins with a white background to form a square input. After editing, we crop out the padded margins according to the recorded padding metadata and resize the edited content back to the original resolution. This geometry-aware procedure keeps the original and edited images pixel-aligned in non-edited regions. As a result, the contrastive supervision is concentrated on the localized semantic change specified by $\mathcal{T}$, rather than on resolution artifacts, aspect-ratio distortion, or unintended background shifts.

**Validity Verification.** Because generative image editing can fail or introduce unintended artifacts, we apply a post-hoc verification loop using QwenVL-Plus. The expert VLM re-evaluates the generated triplet $(m', \mathcal{T}, y'_{new})$ under two criteria.

First, it performs a *visual consistency check* to determine whether the edited image $m'$ faithfully implements the semantic change specified by $\mathcal{T}$. Samples with failed edits, excessive background changes, structural distortion, or unintended modifications to non-target objects are discarded. Second, it performs *response rectification* by checking whether $y'_{new}$ accurately describes the edited image $m'$. Minor textual mismatches or residual hallucinations are corrected through minimal editing, while samples requiring substantial rewriting are removed.

The entire pipeline for processing each sample involves a single call for strategy formulation, response rewriting, visual editing, and verification respectively. Table 5 presents the computational overhead for deploying the visual editor on local NVIDIA H20 GPU. For other stages, we relied on API calls via DashSope.[5]

# C. Implementation Details

All methods are trained on fixed base models with the same training set. Token masks in edited samples are applied to all methods. For methods that do not model symmetrical relations (i.e. DPO, mDPO, V-DPO), we simply split the samples $(m, m', x, y, y')$ into separate image-text pairs $(m, x, y)$ and $(m', x, y')$. The objective of SymMPO (Liu et al., 2025b) includes a term $\mathcal{L}_{DPO_m}$ that requires extra annotation of rejected responses for both the original and contrastive images. To ensure fair comparison, we exclude this term which in practice could be integrated into all methods given extra annotation.

---

[3] https://huggingface.co/Qwen/Qwen-Image-Edit-2511
[4] https://huggingface.co/lightx2v/Qwen-Image-Edit-2511-Lightning
[5] https://dashscope.console.aliyun.com

We follow the paradigm of OPA-DPO (Yang et al., 2025) to first finetune the base models with LoRA (Hu et al., 2022), using the chosen samples from the training data for on-policy alignment. In practice, we merge synthetic and edited chosen samples to form a unified SFT dataset. For IC-VCO, we add multi-image samples to align the policy with our anchor prompt instruction. We set the LoRA rank to 128 and alpha to 256, finetune for one epoch with a learning rate of 2e-5 and a batch size of 128. Then, we use the finetuned policy as the reference policy for preference optimization, using another initialized LoRA adapter with the same configuration for training. The preference optimization uses a learning rate of 5e-6 and a global batch size of 64, and runs for one epoch. All runs are conducted on 8 NVIDIA H20 GPUs.

We follow prior works (Wang et al., 2024a; Xie et al., 2024; Liu et al., 2025b) for hyper-parameters initialization. For all methods, we set $\beta = 0.1$, and set the anchor loss weight $\eta = 1$. For IC-VCO, we set $\lambda_1 = 0.75$, $\lambda_2 = 1.75$, $\eta_1 = \eta_2 = 1$, and $\gamma = 0.3$. For all other hyper-parameters in baselines, we use their reported values.

For evaluation, we use greedy decoding for deterministic prediction, with max_new_tokens=128. For HallusionBench and AMBER, we use Qwen-Flash API via DashScope to map model predictions to yes-or-no labels. For multiple-choice benchmarks including CRPE, R-Bench, and BLINK, we use exact matching for prediction scoring.

