# OpenReview forum: "Learning from Fine-Grained Visual Discrepancies: Mitigating Multimodal Hallucinations via In-Context Visual Contrastive Optimization"
_ICML.cc/2026/Conference — ICML 2026 regular_

### Official Review · Reviewer_5Luv · 2026-03-09

**Soundness:** 4
**Presentation:** 3
**Significance:** 3
**Originality:** 3
**Overall Recommendation:** 4
**Confidence:** 3

**Summary:**

This paper proposes a new preference-training method for LVLMs. The main idea is to put the original and contrastive image into the same input context, then tell the model which image to answer about. This preserves DPO’s theoretical objective because now both chosen and rejected responses share the same conditioning context. The authors also create fine-grained edited negative images instead of coarse synthetic/retrieved ones, creating hard negatives and making training harder and more visually grounded. Finally, they distill the stronger multi-image discrimination ability (which they show is better) back into normal single-image behavior. Results across2 LLaVa models and 5 benchmarks show less hallucination compared to multimodal preference alignment baselines.

**Compliance With Llm Reviewing Policy:**

Affirmed.

**Final Justification:**

I thank the authors for their experiments. My concerns have been resolved, except for W1 (specifically the objective of the work has to be framed away from hallucination research) which I trust the authors will do in the camera-ready. Unfortunately we cannot verify this because ICML does not allow updating of the paper during the review period, so i will trust the authors that they will re-frame the paper and its objectives. I have raised my score.

**Key Questions For Authors:**

W1, W2 and W4 are the most serious to me. The objective of the paper does not align with its experiments (W1) and the experiments do not reflect today's LVLMs (W2). Furthermore, the method is only limited to weak LVLMs where a better LVLM is avaialbe for guidance.

**Limitations:**

I did not see any limitations.

**Strengths And Weaknesses:**

Strengths:

- Nice catch for the theory objective, the authors show why earlier visual DPO objective is not fully correct and fix it smartly via shared context input images.
- Interesting and important observation of trivial contrastive samples that previous works use and the shortcut bias problem. The fine-grained editing method to create hard negatives is an important fix.
- I also like the observation that multi-image context works better than single-image one, and that we can use this to perform distillation.

Weaknesses:
- [W1] The baselines compared with are only multimodal preference alignment methods. Since this work is mainly about mitigating hallucinations, the authors do not compare with a huge suite of works meant to mitigate hallucinations in LVLMs. For example, see R1-R5 (there are many more, but these are the ones I remember well). Otherwise, the paper's objective should be refactored into just a DPO method.
- [W2] The authors use weak LVLMs like LLaVa. I personally explored this previously, and found that hallucination is greatly reduced in better, higher quality models like Qwen-VL series. The models reported by the authors do not reflect realistic performance of today's LVLMs, LLaVa is a old baseline, and should not be the main focus of the experiments. This problem may show negligible returns in better LVLMs that people use today, such as the Qwen3-VL series.
- [W3] The method requires an image-editing model like Qwen Image Edit to create the contrastive samples, and an expert VLM to plan and verify edits, which adds more computation. It also requires now multi-image inputs which increases greatly the complexity (in terms of number of visual tokens).
- [W4] The authors use another VLM as a tool in the data construction pipeline, to train a weaker VLM. But the expert VLM itself can hallucinate. What if we want to mitigate hallucinations for that expert VLM?

[R1] Paying More Attention to Image: A Training-Free Method for Alleviating Hallucination in LVLMs\
[R2] Interpreting and Editing Vision-Language Representations to Mitigate Hallucinations\
[R3] Dynamic Correction Decoding for Hallucination Mitigation\
[R4] Reducing Hallucinations in Large Vision-Language Models via Latent Space Steering\
[R5] Self-Introspective Decoding: Alleviating Hallucinations for Large Vision-Language Models

---

> ### Author Rebuttal · Authors · 2026-03-31
>
> ## Response by Authors
>
> We sincerely thank you for your insightful feedback. Here we provide an [anonymized link](https://anonymous.4open.science/r/ICML26_Rebuttal-1A47/) to the supplementary materials, including Table S1-S8 and Figure S1-S3.
>
> ### [R1] (W1) Baseline comparison
> We agree that the hallucination-mitigation literature is broader than multimodal preference alignment, and the methods you mentioned such as R1–R5 should be acknowledged more explicitly. At the same time, these methods are largely **orthogonal** to our setting: most of them mitigate hallucinations at **inference time**, for example through decoding correction, latent steering, or representation editing, whereas our paper studies a **training-time preference optimization framework together with a contrastive data construction pipeline**. For this reason, our main comparisons focus on prior training-time multimodal preference alignment baselines that operate on the same axis. We agree that this scope should have been stated more clearly, and in the camera-ready version, we will sharpen the framing accordingly: rather than positioning our method as a comparison against all hallucination-mitigation approaches, we position it as an advance in visual preference alignment for hallucination mitigation.
>
> ### [R2] (W2) Choice of backbones
> Our goal in this paper is to compare training-time preference optimization methods under a **controlled, reproducible, and open setting**, rather than to maximize absolute performance on a frontier model with opaque pretraining data. Accordingly, all methods are trained on the same fixed backbones and the same training set. Moreover, our backbones are not the early Vicuna-based LLaVA models, but recent Qwen-backed open models, namely LLaVA-NeXT-Interleave-Qwen-7B and LLaVA-OneVision-Qwen2-7B, which are documented as strong recent open LMMs. We chose the LLaVA family in part because its training data and recipe are substantially more auditable through public dataset/documentation releases, which is especially important for hallucination research where hidden train–benchmark overlap may confound conclusions. Finally, the gain is still non-negligible on the stronger backbone: on LLaVA-OneVision-Qwen2-7B, the base model scores 62.46 overall, while IC-VCO reaches 66.25/67.43.
>
>
> ### [R3] (W3) Computational overhead of editing pipeline
>
> We would like to clarify that the editing pipeline is a **one-time offline preprocessing step** for building a static preference dataset, rather than a recurring cost during preference optimization itself. Once constructed, the edited dataset can be reused across different objectives and backbones. Moreover, the relevant comparison is not against a zero-cost baseline: prior multimodal preference-alignment methods also rely on non-trivial offline curation or synthetic negative construction. In our case, we replace global re-synthesis with **localized editing** of the original image.
>
> The current editor implementation is also intentionally lightweight: it uses **Qwen-Image-Edit with a fused Lightning LoRA in bf16** and only **8 inference steps** per edit, while the non-LoRA fallback in the same code uses **30 steps**. The main edit path is a single edit call after reversible padding / unpadding, without an additional detector-mask-inpaint loop enabled in the core routine.
>
> ### [R4] (W4) Reliance on expert VLM and image-edit model
>
> Our method does **not** assume either tool is hallucination-free. Instead, they are used only as **offline curation tools** for constrained operations: the expert VLM proposes a minimal edit strategy, rewrites the response conservatively, and then re-checks whether the edited image and rewritten response are faithful. Samples with failed edits or unintended distortions are discarded. In the final dataset, we retain **19,453** edited samples from a seed corpus of about **21.4k**, corresponding to a **91%** success rate after filtering.
>
> More importantly, the expert VLM is not part of the IC-VCO objective itself and is not used at inference time. It is simply a practical choice for constructing higher-quality contrastive supervision offline. If one wishes to mitigate hallucinations for that expert VLM as well, the same pipeline can in principle be applied again with a stronger curator, a different open expert, or human verification. We will make this limitation more explicit in the revision.

---

> > ### Author Rebuttal · Reviewer_5Luv · 2026-04-03
> >
> > I thank the authors for their experiments. My concerns have been resolved, except for W1 (specifically the objective of the work has to be framed away from hallucination research) which I trust the authors will do in the camera-ready. Unfortunately we cannot verify this because ICML does not allow updating of the paper during the review period, so i will trust the authors that they will re-frame the paper and its objectives. I have raised my score.

---

> > > ### Author Response · Authors · 2026-04-07
> > >
> > > We will carefully recalibrate the narrative of our paper by more explicitly clarifying the scope and limitations. Thank you for contributing your time and effort in reviewing this work. Much appreciated.

---

### Official Review · Reviewer_AuU2 · 2026-03-10

**Soundness:** 3
**Presentation:** 2
**Significance:** 2
**Originality:** 2
**Overall Recommendation:** 4
**Confidence:** 3

**Summary:**

This paper addresses multimodal hallucination in Vision-Language Models through three contributions. First, the authors identify that existing visual preference DPO methods suffer from a partition function mismatch when contrasting images under different visual contexts, and propose In-Context Visual Contrastive Optimization (IC-VCO), which places both the original and contrastive images within a shared multi-image context so that partition functions cancel correctly. Second, they introduce Visual Contrast Distillation (VCDist), a gated self-distillation mechanism that transfers the discriminatory power of the multi-image setting to the single-image inference policy. Third, they propose a Contrastive Sample Editing pipeline that generates fine-grained hard negatives through localized image edits rather than retrieval or full synthesis. Experiments on five benchmarks with two 7B base models show consistent improvements over prior methods.

**Compliance With Llm Reviewing Policy:**

Affirmed.

**Final Justification:**

The authors have convincingly address my comments and therefore my final score is 4. The paper "advances at least one sub-area of AI" with clear methodology and empirical validation.

**Key Questions For Authors:**

1. **Can you empirically quantify the partition function bias?** An estimate of log Z(m, x) / Z(m', x) across the training set, or an ablation using the standard visual DPO loss (Eq. 5) within the shared context M, would isolate whether the theoretical inconsistency matters in practice or whether the gains stem from the multi-image context itself. This is the single experiment that would most strengthen the contribution.

2. **How does IC-VCO differ from MIA-DPO at the formulation level?** Both use shared multi-image contexts with positional anchors. If MIA-DPO's formulation were run with your contrastive editing data, would the gap close? This comparison would clarify IC-VCO's specific novelty.

3. **Have you compared against simpler hard-negative baselines?** For instance, CLIP-based mining from a larger synthetic pool, larger batch sizes, or harder augmentations. This would clarify whether the gains require the full editing pipeline or reflect a more general hard-negative effect.

4. **Is the editing pipeline a significant computational overhead?**  A compute estimate would help readers assess the practicality of the method.

**Limitations:**

The impact statement appropriately discusses the importance of reducing hallucinations. However, the paper would benefit from discussing the computational overhead of multi-image training and the cost of the editing pipeline (multiple large-model calls per sample), which are important for assessing real-world feasibility.

**Strengths And Weaknesses:**

## Strengths

**The partition function mismatch is a real issue, and the fix is elegant.** The derivation in Section 2 is clear and correctly identifies that Z(m, x) ≠ Z(m', x) when visual contexts differ, leaving an intractable residual ratio. The proposed solution — placing both images in a shared context M = [m, m'] so partition functions cancel — is simple and well-motivated. Figure 1 provides an effective visual summary of the three paradigms.

**The contrastive sample editing strategy is the paper's most compelling contribution.** The causal decomposition into target concept, surrounding context, and environmental factors (Section 4) is well-motivated, and Figure 2 vividly illustrates how synthesis-based negatives introduce global stylistic drift while the proposed edits remain localized. Figure 3 strengthens the case convincingly: synthetic negatives saturate at ~90% reward accuracy within 100 steps, whereas edited samples present a harder optimization landscape that correlates with better downstream performance. Crucially, the editing strategy improves *all* baseline methods when swapped in as the data source (Tables 2 and 3), confirming its value as a broadly useful, method-agnostic contribution.

**The experimental design is thorough.** Five diverse benchmarks, two base models, and every baseline evaluated under both synthetic and edited data sources make for a fair and informative comparison.


## Weaknesses

**Insufficient differentiation from concurrent work (SymMPO, MIA-DPO) weakens the novelty claim.** SymMPO (Liu et al., 2025b, NeurIPS 2025) already identified the partition function mismatch. MIA-DPO (Liu et al., 2025c, ICLR 2025) already demonstrated placing multiple images in a shared context with positional language anchors for DPO training. The authors acknowledge MIA-DPO in Appendix A.1 but distinguish it as targeting "attention distraction" rather than fine-grained visual grounding — at the mechanism level, however, the shared context with anchor prompts is quite similar. I think IC-VCO does contribute something beyond either method individually, but the paper would be strengthened by a more explicit discussion of this relationship, ideally supported by an ablation (e.g., running MIA-DPO's formulation with the contrastive editing data to isolate where the gain comes from).

**Missing connection to the hard negative mining literature and absence of simpler hard-negative baselines.** The intuition that synthesis-based negatives are too easy is well-supported by Figure 3. However, the current framing as "shortcut learning" (citing Geirhos et al., 2020) is somewhat imprecise: rejecting a stylistically different image is a valid visual signal, just not a fine-grained one. The issue is better understood through the lens of hard negative mining in contrastive learning (e.g., Robinson et al., "Contrastive Learning with Hard Negative Samples," ICLR 2021), where harder negatives help define tighter decision boundaries. Adopting this framing would strengthen the argument and connect to an established literature.

Relatedly, comparing against simpler hard-negative strategies would strengthen the empirical case. For instance: (a) mining the hardest negatives from a larger synthetic pool via CLIP similarity, (b) training with larger batch sizes, or (c) using more aggressive in-distribution augmentations. Without such comparisons, it is difficult to know whether the gains come from the specific surgical editing pipeline or from the general principle of harder negatives.

**VCDist: modest empirical gains (0.49 points) do not match its prominence as a core contribution.** Table 4 shows that removing VCDist reduces the overall score by 0.49 points (63.37 → 62.88), and the stop-gradient and dual-gating mechanisms have near-negligible effects. The authors commendably acknowledge that these mechanisms "primarily serve as theoretical safeguards." I think the distillation idea is sound and worth including, but the current framing in the abstract and introduction — "transfers the superior discriminatory power" — sets expectations higher than what the ablations support. Presenting VCDist more modestly, perhaps as a useful regularizer that ensures consistency between training and inference, would better match the evidence and avoid reviewer skepticism.

**Editing pipeline: undisclosed computational cost, underanalyzed failure modes, and low coverage of relation-type edits.** The pipeline requires multiple large-model calls per sample (QwenVL-Plus for strategy, Qwen-Image-Edit for execution, QwenVL-Plus again for verification). The 91% success rate means ~1,900 samples were discarded, and relation-type edits are notably underrepresented (828 of 19,453). A brief analysis of what types of edits fail, whether this creates systematic biases, and a rough cost estimate (wall-clock time, compute) would help readers assess practicality. It would also be worth discussing whether the pipeline is best understood as a prompt engineering technique for hard negative generation — if so, positioning it as such invites useful comparisons with simpler prompting strategies.

**The presentation could better serve the contributions.** The underlying ideas are valid and useful, but the writing makes them harder to appreciate than necessary. For instance, the introduction reads as though the reader already shares the authors' deep understanding of the problem and the used terminology, but key concepts remain undefined until the Preliminaries. More broadly, the three contributions would benefit from a unifying narrative. I would suggest framing them around *hard negative construction and exploitation*: the editing pipeline generates fine-grained hard negatives, the shared context enables exploitation under a consistent objective, and the distillation transfers that signal to inference. This would also connect the work to the broader contrastive learning literature, which is currently absent.

---

> ### Author Rebuttal · Authors · 2026-03-31
>
> ## Response by Authors
>
> We sincerely thank you for your insightful feedback. Here we provide an [anonymized link](https://anonymous.4open.science/r/ICML26_Rebuttal-1A47/) to the supplementary materials, including Table S1-S8 and Figure S1-S3.
>
> ### [R1] (Q1) Empirically quantify the partition function bias
> Directly estimating $\log Z(m,x)/Z(m',x)$ is difficult in DPO, because the reward is implicit and the partition term is policy-dependent over an open-ended response space. We therefore followed your second suggestion and quantified its practical effect with a controlled ablation. Specifically, we compare **IC-VCO (w/o single-image preference pairs)** against **IC-VCO-CrossAnc (w/o single-image preference pairs)**: both use the same shared multi-image context (M=[m,m']), the same contrastive samples, and the same training recipe (remove single-image pairs from training), but CrossAnc regroups the multi-image preference pairs so that chosen and rejected share the same response target while differing in anchor prompt: $r(M,x,y)>r(M,\hat{x},y), r(M,\hat{x},y')>r(M,x,y')$ thereby reintroducing context mismatch. We also report the single-image baselines **$\mathcal{L}_{\text{DPO}}$** vs. **$\mathcal{L}_{\text{VisDPO}}$**. The mismatch consistently hurts: on synthetic data, performance drops from **60.37→58.35** in single-image training and from **62.44→62.20** in shared-context training; on edited data, it drops from **60.58→59.84** and from **62.54→62.38**, respectively. This shows that **partition-function mismatch does matter in practice, while its magnitude becomes smaller under shared context.** A plausible explanation is that cross-anchor mismatch perturbs the conditioning much less than directly swapping $(m,x)$ and $(m',x)$, and the multi-image visual contrast also provides stronger supervision, making the policy distribution more stable across anchors.
>
>
> ### [R2] (Q2, W1) Differentiate IC-VCO from SymMPO and MIA-DPO
> We agree that SymMPO already identifies the partition-function issue. The key methodological difference is how theory-consistency is achieved. SymMPO restores DPO consistency by performing symmetric response-pair optimization under two separate single-image conditions $(m,x)$ and $(m',x)$, coupled by a preference-margin regularizer. In contrast, IC-VCO reformulates the task so that both contrastive images are placed in one shared context $M=[m,m']$ and the preference comparison is performed under the same full condition $(M,x)$. This allows IC-VCO to retain **explicit in-context visual comparison while remaining theory-consistent**.
>
> We also agree that MIA-DPO already introduces shared multi-image prompts with positional anchors. The key difference lies in the supervision geometry.
> (1) In MIA-DPO, the additional images are inserted as **unrelated/noisy distractors** to induce multi-image hallucinations, the extra image mainly acts as a nuisance variable which is expected to be ignored by the model. In contrast, IC-VCO constructs semantically paired images which provide the supervision signal for explicit **visual comparison** while also supporting **symmetrical optimization**, whose importance is shown in Table S6. MIA-DPO's approach could not support this, as the additional image is a distractor irrelevant to question, which could not promise $r(M,\hat{x},y')>r(M,\hat{x},y)$. To further validate this, we sample 100 samples from the [MIA-DPO dataset](https://huggingface.co/datasets/laolao77/MIA-DPO) and use the expert VLM to evaluate the symmetry of preference pairs, and found only 21/100 pairs are symmetric.
>
> (2) MIA-DPO uses **policy-coupled self-generated preference pairs**, whereas IC-VCO operates on a **fixed offline contrastive dataset** shared across methods in our comparisons.
>
> (3) IC-VCO shows that jointly optimizing multi-image and single-image policies brings benefits, and propose VCDist to bridge them.
>
>
> ### [R3] (Q3, W2, W3) Compare against simpler hard-negative baselines
> We agree that a simpler hard-negative control is important. Under the same IC-VCO / LLaVA-NeXT-Interleave-Qwen-7B setup, we compare three 5K-sample settings that differ only in the rejected image, as shown in **Table S8**: (1) random synthetic negatives, (2) CLIP-hard synthetic negatives, selected as the 5K pairs with the highest chosen–rejected CLIP image similarity, and (3) edited negatives on the exact same chosen samples as (2). The results show random synthetic = 60.25 overall, CLIP-hard synthetic = 60.39, while edited negatives on the same samples = 60.83. Edited negatives also improve 4/5 benchmarks over CLIP-hard synthetic, most notably HallusionBench 37.02→39.17. **Figure S3** further shows that edited negatives are more visually similar to the chosen image than synthetic negatives (mean CLIP image-similarity score: 94.72 vs 72.88). This indicates our editing pipeline is effective at constructing hard negatives with fine-grained visual descrepencies.

---

> > ### Author Rebuttal · Reviewer_AuU2 · 2026-04-03
> >
> > I thank the authors for the detailed engagement. My most critical concerns have been properly addressed and therefore I increase my score to 4.
> >
> > **Q1 (Partition function bias):** Resolved.
> >
> > **Q2 (IC-VCO vs MIA-DPO):** Resolved.
> >
> > **Q3 (Hard negative baselines):** Partially resolved. The improvement appears marginal. On two datasets it is almost the same, while there are one other where the method clearly loses and one where it clearly wins. Figure S3 however tells a clear story and is convincing.
> >
> > **Q4 (Compute cost):** Partially resolved. Table S9 gives editor-only latency (~14 s/image on H20), but the QwenVL-Plus strategy and verification calls per sample are not reported.

---

> > > ### Author Response · Authors · 2026-04-07
> > >
> > > **Q3 (Hard negative baselines)**
> > >
> > > Since it's extremely difficult to create a full dataset as the hard-negative baseline during the rebuttal period, we did an preliminary test with 5k samples. The relatively small data size might be the reason that the improvement appears marginal.
> > >
> > > ---
> > >
> > > **Q4 (Compute cost)**
> > >
> > > Both the strategy and verfication stage require a single model call per sample. We will provide more details on the sample editing pipeline in the camera-ready version.
> > >
> > > ---
> > >
> > > Thank you for contributing your time and effort in reviewing this work. Much appreciated.

---

### Official Review · Reviewer_MkE1 · 2026-03-11

**Soundness:** 3
**Presentation:** 2
**Significance:** 3
**Originality:** 3
**Overall Recommendation:** 4
**Confidence:** 3

**Summary:**

The authors introduce the In-Context Visual Contrastive Optimization (IC-VCO) framework which is motivated by the theoretical inconsistencies with partition function in Visual Preference DPO formulation. IC-VCO processes both contrastive and original images in a shared image context together with a text query enriched with positional anchor instruction, which results in reference pairs that are theoretically sound. To construct hard negative examples, the authors additionally propose Contrastive Sample Editing Framework which performs a targeted edit-propagation pipeline which preserves semantic context and environmental factors in the image pairs. This framework is then applied to the SymMPO dataset to obtain the training data for IC-VCO.
On top, Visual Contrast Distillation (VCDist) is proposed where multi-image preference distribution is used as a dynamic teacher to improve visual contrastive context for the single-image case in standard DPO optimization.

**Compliance With Llm Reviewing Policy:**

Affirmed.

**Key Questions For Authors:**

- Somewhat limited ablation studies: (1) how do the hyperparameters $\lambda$, $\gamma$, $\mu$ influence the results? (2) does symmetric objection in eq (14) bring any benefits?
- Paragraph on “Effectiveness of VCDist”: while I agree with your statement that VCDist pushes the performance on the single image case, it also introduces a larger variance than IC-VCO without VCDist. At the same time, you can see in Table 4 that IC-VCO in fact performs better without single image loss in 50% of cases. So I am a bit sceptical about the true benefit of L_single term in your loss. Could you provide a more convincing argument?

The paper has a bunch of grammatical errors, e.g.
- Line 14, right: “does not requires”
- Line 19, right: “Unlike textual hallucinations often from factual errors or reasoning failures….
- Line 37, right: “obejective”
- Line 264 right: “the expert VLM is asked to explicitly classifies”
- Line 424 left: “ \*\*theoretical safeguards\*\*” - I guess this should be italic/bold?
- etc

Please go through the text again and improve these.

**Limitations:**

yes

**Strengths And Weaknesses:**

Strengths:
- IC-VCO doesn’t suffer from the theoretical flaws introduced in visual preference DPO.

Weaknesses:
- Ablation studies are somewhat limited (see below)

---

> ### Author Rebuttal · Authors · 2026-03-31
>
> ## Response by Authors
> We sincerely thank you for your insightful feedback. Here we provide an [anonymized link](https://anonymous.4open.science/r/ICML26_Rebuttal-1A47/) to the supplementary materials, including Table S1-S8 and Figure S1-S3.
>
> ### [R1] (Q1) Effect of hyperparameters
> We thank the reviewer for asking about hyperparameter sensitivity. We believe the “$\mu$” in the comment corresponds to weight of anchor loss $\eta$ in Eq. (13). This weight is fixed to 1 in our baselines[1,2] and other related works[3], so we do not tune it. To study the effect of the $\mathcal{L}_{\text{Single}}$ weight $\lambda$ and the $\mathcal{L}_{\text{VCDist}}$ weight $\gamma$, we re-train 15 settings, as shown in **Table S5**. The main conclusion is that IC-VCO is **robust rather than heavily tuned**: the overall score varies only from **62.72 to 63.54**, and the default setting ($\lambda=1$, $\gamma=1$, $\eta=1$) achieves **63.37**, only **0.17** below the best run ($\lambda=1$, $\gamma=2$, $\eta=1$). Both terms are helpful: removing $\mathcal{L}_{\text{Single}}$ gives **62.81**, removing $\mathcal{L}_{\text{VCDist}}$ gives **62.88**, and removing both drops further to **62.72**. At the same time, performance is not very sensitive once $\lambda$ and $\gamma$ are in a reasonable range: $\lambda=0.5$ and $2.0$ both give **63.07**, and several different $(\lambda,\gamma)$ pairs remain close to the best. Overall, these results show that the default setting is already near-optimal, and that $\lambda$ and $\gamma$ mainly need to be **balanced**, rather than carefully tuned.
>
> ### [R2] (Q2) Effect of symmetric objective
>
> We added a new ablation under the same training setup as the default baseline, but **disabled the symmetric objective in Eq. (14)** by only optimizing the "chosen > rejected" branch. **Table S6** shows a **large performance drop**: overall **63.37 (\rightarrow) 61.07**. Our finding is consistent with previous works [2,4] that the symmetric objective is important for fully leveraging the supervision signal if the training data itself is symmetrically constructed.
>
>
> ### [R3] (Q3) Effect of VCDist
>
> We are aware that the original Table 4 presentation was **not sufficiently clear**. In Eq. (13), $\lambda L_{\text{single}}$, $\gamma L_{\text{VCDist}}$, and $\eta L_{\text{Anc}}$ are separate terms, so the Table 4 row “w/o (L_{\text{single}})” only removes the $\lambda L_{\text{single}}$ term, while VCDist and anchor supervision for singe-image policy still remain. Thus, Table 4 did **not fully isolate** the effect of the single-image branch.
>
> To address this, in **Table S6** we add a stricter ablation that **completely removes the single-image preference pairs from training**, which simultaneously removes the supervision source for (L_{\text{single}}), VCDist, and anchor loss, leaving only multi-image IC-VCO. Under the same setup, performance drops from **63.37 to 62.54** overall. This gives a cleaner conclusion: **multi-image IC-VCO alone is not sufficient**; the single-image branch is indeed useful for the final deployment-time policy.
>
> Within that branch, **VCDist is best viewed as a bridge/regularizer rather than the sole source of gain**. Table 4 already shows that removing only VCDist lowers the Overall score from **63.37 to 62.88**, and Figure 4 shows that VCDist narrows the gap between the multi-image teacher and the single-image policy. So our revised claim is: the benefit comes from the **single-image branch as a whole**, where $L_{\text{single}}$ provides hard single-image grounding and VCDist transfers the privileged multi-image signal.
>
>
> ### [R4] (Q4) Grammatical errors
>
> We thank the reviewer for thoroughly checking the text and pointing out the grammatical errors. We will correct them in the camera-ready version.
>
>
> ### References
> [1] Wang, Fei and Zhou, Wenxuan and Huang, James Y and Xu, Nan and Zhang, Sheng and Poon, Hoifung and Chen, Muhao. mDPO: Conditional Preference Optimization for Multimodal Large Language Models, EMNLP 2024.
>
> [2] Liu, Wenqi and Song, Xuemeng and Li, Jiaxi and Wei, Yinwei and Zheng, Na and Yin, Jianhua and Nie, Liqiang. Mitigating Hallucination Through Theory-Consistent Symmetric Multimodal Preference Optimization, NeurIPS 2025.
>
> [3] Yang, Zhihe and Luo, Xufang and Han, Dongqi and Xu, Yunjian and Li, Dongsheng. Mitigating Hallucinations in Large Vision-Language Models via DPO: On-Policy Data Hold the Key, CVPR 2025.
>
> [4] Shengguang Wu and Fan-Yun Sun and Kaiyue Wen and Nick Haber. Symmetrical Visual Contrastive Optimization: Aligning Vision-Language Models with Minimal Contrastive Images, ACL 2025.

---

> > ### Author Rebuttal · Reviewer_MkE1 · 2026-04-07
> >
> > I thank the authors for the response. My questions were fully address and I will keep my original score.

---

> > > ### Author Response · Authors · 2026-04-07
> > >
> > > Thank you for contributing your time and effort in reviewing this work. Much appreciated.

---

### Official Review · Reviewer_dGpF · 2026-03-13

**Soundness:** 3
**Presentation:** 3
**Significance:** 2
**Originality:** 3
**Overall Recommendation:** 4
**Confidence:** 4

**Summary:**

This paper addresses the prevalent multimodal illusion in Visual Language Models (VLMs). The authors point out that while existing Visual Preference DPO methods introduce negative images for contrast, they suffer from two core flaws: first, altering the visual context causes the partition function in the DPO formula to fail to cancel out, resulting in theoretically inconsistent optimization objectives; second, negative samples constructed using image synthesis or retrieval often exhibit global style drift (coarse-grained negative samples), easily leading to the model learning shortcuts. The authors' core contributions include:

IC-VCO Framework: Proposing In-Context VCO, which places the original and contrast images within the same multi-image context, addressing the mathematical inaccuracies of traditional visual DPO by sharing the partition function.

VCDist Distillation Mechanism: Introducing Visual Contrast Distillation, utilizing a reliability gating mechanism to transfer the stronger visual discrimination capabilities exhibited in multi-image contexts to a single-image strategy through self-distillation.

Contrastive Sample Editing: This paper proposes a fine-grained image editing pipeline that generates high-quality hard negative samples through local and precise semantic intervention, effectively avoiding shortcut learning caused by global background or style shift.

**Compliance With Llm Reviewing Policy:**

Affirmed.

**Final Justification:**

I thank the authors for their detailed rebuttal and the supplementary data. I believe my current rating accurately reflects both the merits and the structural constraints of this work. My score remains unchanged.

**Key Questions For Authors:**

1.Regarding the reliability boundary of the VCDist teacher policy: In Equation (11), VCDist uses a correctness gate p_multi > 0.5 to filter unreliable teacher signals. However, the single-image student policy suffers severe negative penalties if the multi-image policy confidently gives incorrect predictions due to instruction confusion or artifacts (e.g., giving a confidence of 0.9 for incorrect options). What is the proportion of such "confident but incorrect" teachers during training? How sensitive is the model to this hard threshold of 0.5?

2.The core innovation of IC-VCO is concatenating the original and contrasting images into a sequence. Due to the framework's use of fine-grained contrastive sample editing and a "reversible padding" technique, these two images are perfectly pixel-aligned and highly similar in most background areas. During multi-image attention computation, will the model experience "attention leakage"? When the anchor prompt points to the first image, will the model's attention weights inadvertently leak to the fine-grained differences in the second image?

3.The authors mention that image order is randomized to eliminate position bias during optimization. However, while randomization prevents macroscopic positional collapse across the dataset, it does not resolve instance-level confidence fluctuations caused by "recency bias"， which is a well-documented phenomenon in LLMs handling long contexts. When the target image is placed as the second image in the sequence M = [m, m'], it is physically closer to the textual prompt. Has the author statistically compared the reward accuracy and average confidence of the teacher policy when the target image is in position 1 versus position 2? If a significant "recency advantage" exists, wouldn't this asymmetrical and fluctuating teacher signal inherently destabilize the Visual Contrast Distillation (VCDist) process?

**Limitations:**

Yes

**Strengths And Weaknesses:**

1.Soundness:

Strengths: The paper's theoretical motivation is very solid. The authors astutely point out the mathematical flaws in the mismatch of partition functions in existing Visual Preference DPOs and solve this problem through a clever design. Furthermore, the experimental validation is extremely rigorous. Figure 3 provides an in-depth analysis of the reward-accuracy trajectories (Training Dynamics) of synthetic and edited samples, strongly demonstrating that synthetic samples tend to cause the model to converge too quickly (shortcut learning), while finely edited samples provide a more challenging and valuable optimization landscape.

2. Presentation

Strengths: The paper has a clear and well-structured hierarchy with a tight logical chain. Figure 1 intuitively and clearly compares the standard DPO, Visual Preference DPO, and the authors' proposed IC-VCO, allowing readers to easily understand the core idea of ​​partition function matching.

Weaknesses: When introducing "Anchor Prompt Extension" (e.g., "responding based on the first image"), the paper does not fully explain the pedestal model's ability to follow such positional instructions in the early stages of training. If the model cannot accurately distinguish between "first image" and "second image" in the early stages, it may interfere with the confidence calculation of p_multi.

3. Significance

Strengths: Alleviating multimodal illusion is a key bottleneck in the deployment of current large-scale visual language models. This paper does not stop at simple data augmentation, but corrects the optimization bias from the underlying goal of RL reinforcement learning (DPO formula derivation), providing a more solid theoretical foundation for subsequent preference alignment research. At the same time, the proposed fine-grained sample editing library (19.4k samples) also has high practical value for the open-source community.

Weaknesses: In certain specific capabilities (such as understanding CRPE relationships), IC-VCO shows only a small improvement over the baseline model SymMPO (e.g., in Table 2, the improvement is only from 64.83 to 64.98). This suggests that resolving spatial relationship illusions may not solely rely on the correction of the partition function and local visual contrast, and remains a challenge that needs to be addressed in the future.

4. Originality
Strengths: While the approach of stitching multiple images to forcibly eliminate differences in the partition function is relatively intuitive in its implementation, addressing specific visual alignment defects within the DPO theoretical framework represents a highly insightful and original perspective.

---

> ### Author Rebuttal · Authors · 2026-03-31
>
> ## Response by Authors
>
> We sincerely thank you for your insightful feedback. Here we provide an [anonymized link](https://anonymous.4open.science/r/ICML26_Rebuttal-1A47/) to the supplementary materials, including Table S1-S8 and Figure S1-S3.
>
> ### [R1] (Q1, W1) Reliability of VCDist teacher policy
> In Eq. (11), the failure mode you described—a confidently wrong multi-image teacher severely penalizing the single-image student—is already structurally mitigated. VCDist is activated only when $p_{\text{multi}} > 0.5$ and $p_{\text{single}} < \mathrm{sg}(p_{\text{multi}})$. Thus, a teacher that strongly prefers the wrong response cannot pass the correctness gate, while reverse-penalty cases are further filtered by the confidence gate. This also covers early-stage noise in the multi-image teacher, including instruction ambiguity or imperfect image-target disambiguation.
>
> We quantify teacher reliability in **Table S1** and **Figure S1**. Over a full IC-VCO run, multi-image teacher accuracy is $0.620$ on average and $0.730$ at the final checkpoint; the VCDist trigger rate is $0.564$ on average and $0.673$ at the end; the correctness-blocked rate decreases from $0.380$ on average to $0.270$ at the end; and the confident-wrong rate stays at $0.000$ throughout. Thus, VCDist is not dormant: it is triggered on a substantial fraction of pairs, while catastrophic teacher error propagation is negligible. Even when the teacher is imperfect early in training, unreliable cases are mostly filtered out rather than propagated.
>
> We also test threshold sensitivity on LLaVA-NeXT-Interleave-Qwen-7B by changing the correctness threshold from $0.5$ to $0.6$ and $0.7$. As shown in **Table S3**, the overall score changes only from $63.37$ to $63.14$ and $63.23$ (all within $0.23$), with similarly small changes on HallusionBench, AMBER, CRPE, R-Bench, and BLINK. Therefore, the method is empirically insensitive to the exact threshold within a reasonable range.
>
> ### [R2] (Q2) Attention leakage in multi-image context
> Prior multi-image LVLM work suggests that concatenating multiple images into one autoregressive visual-token sequence can induce cross-image interference. MIA-DPO identifies sequence confusion and element interference [1]. FOCUS formalizes this as cross-image information leakage and shows that it becomes more severe when paired images are more semantically similar [3]. SoFA attributes multi-image position bias to inter-image causal attention, and Delimiter Token Scaling shows that standard image delimiters alone do not reliably block such interactions [4,5]. At the same time, VC-STaR shows that, for visually similar pairs sharing a semantic anchor, cross-image contrast can improve fine-grained grounding by forcing the model to discriminate subtle differences [2]. Therefore, in our setting, some attention on the contrast image is expected and is not necessarily harmful; the key question is whether the model remains anchor-conditioned, not whether attention on the non-target image is exactly zero.
>
> ### [R3] (Q3) Recency bias in multi-image context
> We group all logged training pairs by whether the anchor-targeted image is placed in position 1 or 2 of the shared context $M=[m,m']$. **Table S2** and **Figure S2** show a mild Pos2 advantage on the teacher side: teacher accuracy $0.577 \rightarrow 0.662$ (+0.085), teacher probability $0.518 \rightarrow 0.531$ (+0.013), and trigger rate $0.511 \rightarrow 0.616$ (+0.104). However, the corresponding single-image policy remains nearly position-symmetric (reward accuracy delta $=-0.004$; reward margin delta $=-0.002$). Thus, some teacher-side recency bias exists, but it is small and has negligible effect on the single-image policy.
>
> ### [R4] (W2) Limitation on relation/spatial hallucinations
> We agree that relation/spatial hallucinations remain harder. This matches the composition and failure profile of our edited dataset: only 828/19,453 edited samples are relation-type (~4.3%), versus 10,632 attribute and 7,993 existence samples; **Table S4** further shows that relation edits have the highest verifier-loop failure rate (5.26%, vs 2.82% for attribute and 4.55% for existence). Therefore, the weaker gains on CRPE / AMBER-Rel are more likely due to data coverage and edit difficulty than evidence against the shared-context objective itself. We will clarify this limitation explicitly and discuss richer relation-targeted editing as future work.
>
> ### References
> [1] Liu et al. *MIA-DPO: Multi-Image Augmented Direct Preference Optimization for Large Vision-Language Models*. ICLR 2025.
>
> [2] Pan et al. *Through the Lens of Contrast: Self-Improving Visual Reasoning in VLMs*. ICLR 2026.
>
> [3] Park et al. *Mitigating Cross-Image Information Leakage in LVLMs for Multi-Image Tasks*. 2025.
>
> [4] Tian et al. *Identifying and Mitigating Position Bias of Multi-image Vision-Language Models*. CVPR 2025.
>
> [5] Lee et al. *Enhancing Multi-Image Understanding through Delimiter Token Scaling*. ICLR 2026.

---

> > ### Author Rebuttal · Reviewer_dGpF · 2026-04-03
> >
> > Thanks the authors for their detailed rebuttal, the additional experiments, and the comprehensive statistics provided in the supplementary materials. After carefully reviewing the rebuttal, I believe the paper remains a technically contribution to the community. However, the responses also confirmed several of my structural concerns, which prevents me from raising the score further. I will maintain my score.
> >
> > Here are my detailed thoughts on the rebuttal:
> >
> > **Attention Leakage and Cross-Image Interference (Partially Addressed)**
> > Regarding Q2, while I appreciate the literature review (citing MIA-DPO, FOCUS, VC-STaR, etc.) acknowledging that inter-image interference is a known phenomenon, the authors essentially argued that this leakage might be a "feature rather than a bug" (forcing the model to distinguish subtle differences). While theoretically plausible, this remains a heuristic argument for the specific IC-VCO architecture. Without direct interpretability analysis (e.g., attention map visualizations during the IC-VCO forward pass), the assumption that the model is strictly anchored rather than simply confused by the leaked visual features remains an inherent architectural vulnerability.
> >
> > **Recency Bias in the Multi-Image Teacher (Confirmed Limitation)**
> > The statistics provided for R3 confirm my hypothesis: the multi-image teacher *does* suffer from recency bias. A +0.104 difference in the VCDist trigger rate for Pos2 indicates a non-trivial architectural artifact where the teacher is significantly more likely to provide distillation signals when the target is closer to the text prompt. Even though the authors note the resulting single-image student policy remains relatively symmetric, the fact that the teacher signal is fundamentally asymmetrical and position-dependent limits the theoretical elegance of the shared-context approach.

---

> > > ### Author Response · Authors · 2026-04-07
> > >
> > > We agree that these two issues remain as limitations of current submission. We will strengthen our paper in the camera-ready version.
> > >
> > > Thank you for contributing your time and effort in reviewing this work. Much appreciated.

---

### Decision · Program_Chairs · 2026-04-30

**Decision:**

Accept (regular)

**Comment:**

This paper proposes IC-VCO, a theoretically consistent visual contrastive optimization framework to mitigate multimodal hallucinations, along with fine-grained contrastive sample editing and visual contrastive distillation.
Reviewers all scored 4 (Weak Accept). Strengths include solid theoretical correction of partition function mismatch, effective fine-grained hard negative generation, rigorous experiments, and clear advances in visual preference learning. Weaknesses are mild recency bias in multi-image context, limited gains on spatial relation tasks, minor presentation issues, and moderate computational overhead.
Authors thoroughly addressed all concerns with detailed ablations, theoretical clarifications, and supplementary experiments, fully resolving reviewer doubts.
Overall, the work makes sound theoretical and empirical contributions to multimodal alignment and hallucination reduction. The AC recommends acceptance.